# The plant unique ESCRT component FREE1 regulates autophagosome closure

Yonglun Zeng[1,9], Baiying Li[1,9], Shuxian Huang[1,9], Hongbo Li[2], Wenhan Cao [1], Yixuan Chen[1], Guoyong Liu [3], Zhenping Li[1], Chao Yang[2], Lei Feng [1], Jiayang Gao[1], Sze Wan Lo[1], Jierui Zhao[4,5], Jinbo Shen [6], Yan Guo [3], Caiji Gao[2], Yasin Dagdas [5] & Liwen Jiang [1,7,8] ✉

The energy sensor AMP-activated protein kinase (AMPK) can activate autophagy when cellular energy production becomes compromised. However, the degree to which nutrient sensing impinges on the autophagosome closure remains unknown. Here, we provide the mechanism underlying a plant unique protein FREE1, upon autophagy-induced SnRK1α1-mediated phosphorylation, functions as a linkage between ATG conjugation system and ESCRT machinery to regulate the autophagosome closure upon nutrient deprivation. Using high-resolution microscopy, 3D-electron tomography, and protease protection assay, we showed that unclosed autophagosomes accumulated in *free1* mutants. Proteomic, cellular and biochemical analysis revealed the mechanistic connection between FREE1 and the ATG conjugation system/ESCRT-III complex in regulating autophagosome closure. Mass spectrometry analysis showed that the evolutionary conserved plant energy sensor SnRK1α1 phosphorylates FREE1 and recruits it to the autophagosomes to promote closure. Mutagenesis of the phosphorylation site on FREE1 caused the autophagosome closure failure. Our findings unveil how cellular energy sensing pathways regulate autophagosome closure to maintain cellular homeostasis.

Autophagy is an evolutionary conserved process that involves the formation of a double-membrane structure called the autophagosome. Dysregulation of autophagy causes a variety of human diseases, including cancer, neurodegeneration, aging, and pathogen infections. In plants, autophagy plays important roles under adverse environments, likewise nutrient starvation, drought, salt and heat[1]. Genetic studies in yeast have revealed the fundamental roles of autophagy-related (*ATG*) genes in autophagosome biogenesis[2-4]. ATG machinery consists of several functional units: Atg1/ULK1 kinase complex, the ATG14-containing PtdIns 3-kinase complex, the Atg9 vesicle, the Atg2-Atg18 complex, the ATG12 conjugation system, and the ATG8 conjugation system[3,5]. The regulatory roles of the evolutionary conserved energy sensor AMPK on triggering the autophagosome initiation through phosphorylation and activation of ULK1 (unc-51-like autophagy-activating kinase 1)[6-8] and differential regulation of VPS34 (vacuolar protein sorting 34) containing complexes[9] have been unveiled. Albeit how nutrient elicits the autophagosome initiation has been revealed, the degree to which

[1]Centre for Cell & Developmental Biology and State Key Laboratory of Agrobiotechnology, School of Life Sciences, The Chinese University of Hong Kong, Hong Kong, China. [2]Guangdong Provincial Key Laboratory of Biotechnology for Plant Development, School of Life Sciences, South China Normal University, Guangzhou, China. [3]State Key Laboratory of Plant Physiology and Biochemistry, College of Biological Sciences, China Agricultural University, Beijing, China. [4]Vienna BioCenter PhD Program, Doctoral School of the University at Vienna and Medical University of Vienna, Vienna, Austria. [5]Gregor Mendel Institute, Austrian Academy of Sciences, Vienna BioCenter, Vienna, Austria. [6]State Key Laboratory of Subtropical Silviculture, Zhejiang A&F University, Hangzhou, China. [7]CUHK Shenzhen Research Institute, Shenzhen, China. [8]Institute of Plant Molecular Biology and Agricultural Biotechnology, The Chinese University of Hong Kong, Hong Kong, China. [9]These authors contributed equally: Yonglun Zeng, Baiying Li, Shuxian Huang. ✉e-mail: ljiang@cuhk.edu.hk

nutrient sensing impinges on the autophagosome closure remains unknown[10].

The endosomal sorting complexes required for transport (ESCRT) is an evolutionarily conserved, multi-subunit membrane remodeling machinery that can form membrane constrictions in a reverse-topology manner. ESCRT proteins play canonical roles in the biogenesis of multivesicular body (MVB) and the formation of intraluminal vesicle (ILV) that formed inside the MVB by invagination and budding of membrane into the lumen of the endosome[11–13]. The core ESCRT machinery consists of ESCRT-0, ESCRT-I, ESCRT-II and ESCRT-III, VPS4 (vacuolar protein sorting associated 4), LIP5 (lysosomal trafficking regulator interacting protein-5), as well as ALIX (ALG2-interacting protein X)[14]. During ILV formation and MVB biogenesis, the ESCRT machinery assembles in a hierarchical fashion on endosomal membranes. ESCRT-0, the best-characterized complex in MVB biogenesis, acts most upstream and serves as an endosomal clathrin adaptor that recruits ESCRT-I complex. The ESCRT-I complex engages ESCRT-II, which in turn nucleates ESCRT-III polymerization through VPS20 activation. The hexameric AAA+ATPase VPS4 then coincides with LIP5 and ultimately disassembles the ESCRT-III complex to recycle its subunits. In eukaryotes, increasing evidence points to the divergent function of ESCRTs in many cellular processes, including cytokinetic abscission[15], HIV budding[16,17], RNA virus replication[18,19], exosome formation[20], defective nuclear pore complexes clearance[21–24], plasma-membrane (PM) repair[25,26], lysosome repair[27,28], and microautophagy[29,30]. Notably, recent studies indicate that ESCRT complexes also function in autophagosome closure[31–34]. In mammal, the ESCRT-I component VPS37A subunit is required for autophagosome closure, followed by recruitment of the ESCRT-III component CHMP2A[31,32]. In yeast, the ESCRT-III subunit SNF7 interacts with the scaffold protein Atg17 in a VPS21-dependent manner, resulting in the recruitment of ESCRT-III for autophagosome sealing[33,34]. In plants, the bona fide roles of ESCRT in the autophagosome closure remain elusive. Furthermore, how ESCRT complex relocates from MVBs to autophagosomes and regulates autophagosome closure remain largely unknown.

Plants encode most ESCRT isoforms in their genome, including ESCRT-I, ESCRT-II, ESCRT-III, and VPS4/SKD1, with the exception of the canonical ESCRT-0 subunits[35,36]. Accumulating evidence demonstrated the conserved and bona fide functions of plant ESCRTs in endosomal sorting, ILV formation and MVB biogenesis[37–40]. Furthermore, plant ESCRT components are involved in a variety of cellular events including cytokinesis, viral replication, chloroplast turnover, autophagic degradation, hormone signalling, biotic and abiotic stress responses[18,19,41–47]. Intriguingly, plants seem to compensate the missing ESCRT-0 subunit with a plant unique ESCRT protein FREE1 (FYVE DOMAIN PROTEIN REQUIRED FOR ENDOSOMAL SORTING 1). FREE1 works in concert with ESCRT-I to regulate endosomal sorting and ILV/MVB/Vacuole biogenesis[48]. Interestingly, our previous studies have shown that under autophagy inducing conditions, substantial amounts of autophagosomal structures accumulated in the cytoplasm of *Arabidopsis free1* mutants[49]. However, the underlying mechanism remains unknown.

In this study, we demonstrated a mechanistic role of FREE1 in regulating autophagosome closure. Using a combination of biochemical methods, live-cell imaging, high-resolution deconvolution and 3D-electron tomography (ET) analysis, we showed that the autophagosomal structures that accumulated in the *Arabidopsis free1* mutants were largely unclosed autophagosomes in nature. Analysis of the FREE1 interactome during nutrient starvation revealed ESCRT-III machinery and ATG12-ATG5-ATG16 complex as FREE1 interactors. Biochemical analyses further proved that FREE1 forms a complex with ESCRT-III and ATG12-ATG5-ATG16 conjugation system, and directly interacts with the conjugation system substrates the ubiquitin-like proteins ATG8a and ATG8i through a classical AIM (Atg8-interacting

motif). Consistently, upon autophagy induction, unsealed autophagosomes were accumulated in the cytoplasm of ATG5 and ESCRT-III mutants. Lastly, immunoprecipitation and mass-spectrum analysis unveiled that FREE1 can be phosphorylated upon nutrient starvation. Interaction and phosphorylation assays demonstrated that KIN10 (SnRK1α1) can directly interact and phosphorylate FREE1. Mutagenesis of the KIN10-targeted phosphorylation site on FREE1 caused the failure of autophagosome closure and defective of plant growth upon starvation. Taken together, during nutrient starvation in plant, the energy sensor SnRK1α1 phosphorylates FREE1 to promote the recruitment of FREE1 to phagophores, and subsequently links the ATG conjugation machinery with ESCRT-III complexes to mediate autophagosome closure. Using the advanced imaging techniques in combination with biochemical methods and genetic approaches, our study comprehensively unveiled the mechanism underlying how nutrient regulates the recruitment of ESCRT machinery to the closing phagophore for achieving autophagosome closure.

## Results
### FREE1 is required for autophagosome closure
Our previous studies have shown that under autophagy inducing conditions, substantial amounts of autophagosomal structures accumulated in the cytoplasm of the homozygous T-DNA insertional mutants *free1* (*free1-/-*) as well as the dexamethasone (DEX)-inducible FREE1 RNAi mutants (*DEX::RNAi-FREE1*)[49]. To elucidate the possible mechanism underlying autophagosome accumulation in FREE1 mutants, we performed live-cell imaging analysis with deconvolution on transgenic *Arabidopsis* plants expressing YFP-ATG8e in *DEX::RNAi-FREE1*[49]. Upon DEX treatment, the expression level of FREE1 was sufficiently knocked-down in the *DEX::RNAi-FREE1* transgenic plants, while the control DEX treatment exhibited little effect on autophagosome formation in plants (Supplementary Fig. 1a, b and Supplementary Fig. 2a, b). Consistent with our previous study[49], YFP-ATG8e labelled autophagosomes accumulated in *DEX::RNAi-FREE1* mutants upon DEX and autophagic induction by benzothiadiazole (BTH) or BTH and Concanamycin A (ConcA) treatments in comparison to mock (Fig. 1a, b and Supplementary Fig. 2c). Intriguingly, substantial amounts of unclosed autophagosomal structures were observed upon BTH-induced autophagy, while the effects were even more pronounced upon BTH and ConcA treatments (Fig. 1a, b). These results are consistent with the previous findings showing that the specific V-ATPase inhibitor ConcA which blocks autophagic/vacuole degradation may further induce the autophagosome formation in the cytoplasm upon autophagy-induced conditions[50–53], albeit the underlying mechanism remains elusive. Live-cell imaging analysis on the dynamics of ATG8e-positive punctae further showed that the autophagosomes in *DEX::RNAi-FREE1* mutants with DEX treatments were unable to close as compared to *DEX::RNAi-FREE1* mutants without DEX treatments upon autophagic induction (Supplementary Fig. 3, Supplementary Movie. 1, and Supplementary Movie. 2). Biochemical analysis further showed that the lipidated ATG8 and the autophagy receptor NBR1 significantly accumulated in the *DEX::RNAi-FREE1* mutants upon DEX induction with or without BTH treatment, suggesting that the autophagosome initiation and progression remained normal in the FREE1 mutants (Supplementary Fig. 4a, b). Nonetheless, the vacuolar turnover assay showed that the YFP-ATG8 turnover was significantly impeded in *DEX::RNAi-FREE1* mutants upon DEX and autophagy induction, indicating that the unclosed autophagosomes were accumulated in the cytoplasm (Supplementary Fig. 4c). To demonstrate that FREE1 depletion indeed causes the accumulation of unsealed autophagosomes biochemically, we next performed the protease protection assay according to the protocol recently developed in plants (Supplementary Fig. 5a)[54]. Our results showed that the autophagic-receptor NBR1 and ATG8 substantially accumulated in the FREE1 depletion mutants upon DEX treatment (Supplementary Fig. 5b). Notably, the

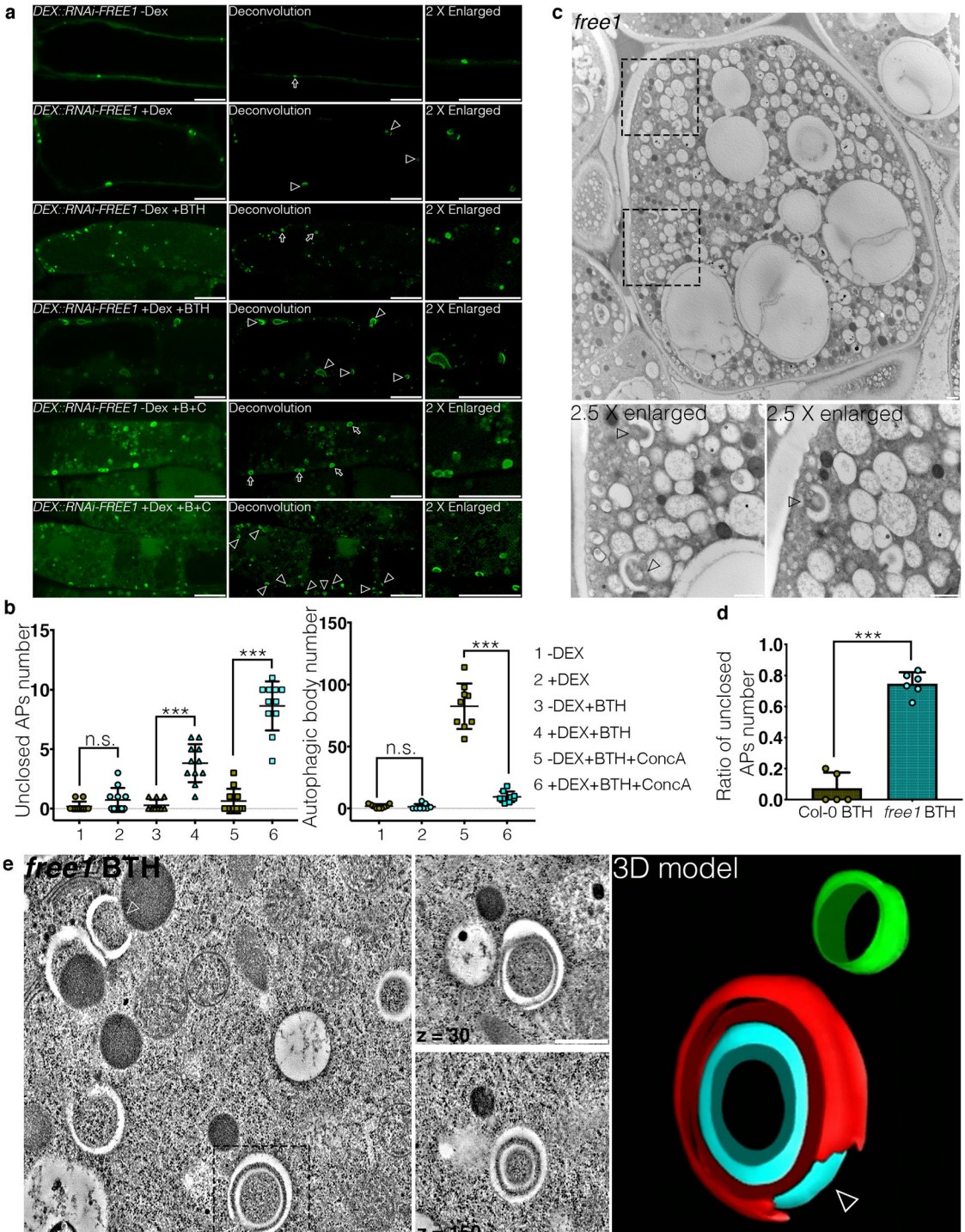

**Fig. 1 | FREE1 mutants accumulate unclosed autophagosomes. a** Transgenic *Arabidopsis* seedlings expressing YFP-ATG8e in dexamethasone-inducible FREE1 RNAi (*DEX::RNAi-FREE1*) mutants were subjected to mock or autophagy inductions by BTH (Benzothiadiazole, B) or BTH and ConcA (Concanamycin A, C) with or without DEX treatment for 8hrs before confocal laser scanning microscopy (CLSM) observation. Images were further processed by Leica Huygens deconvolution. Arrows and arrowheads indicate the close and unsealed autophagosomal structures, respectively. Scale bars, 10 μm. **b** Quantification analysis of the numbers of unclosed autophagosomal structures in the cytoplasm (left panel) and autophagic bodies inside the vacuoles (right panel) per cross-cell section shown in **a**. Means ±SD; $n = 11$ (left panel) and $n = 9$ (right panel) individual cells per experimental group, one-way analysis of variance (ANOVA), followed by Tukey's multiple test; [***]$p < 0.001$; n.s., not significant. **c** TEM (transmission electron microscopy) analysis of high-pressure freezing/frozen substituted homozygous *free1* T-DNA insertional mutant root tips upon BTH treatment for 8 h. Black dash boxes indicate the 2.5 X zoom-in areas. Arrowheads indicate examples of the unsealed autophagosomal structures. Scale bars, 500 nm. **d** Quantification analysis of the ratio of the unsealed autophagosomes shown in **c**. Means±SD; $n = 5$ (Col-0 BTH) and $n = 6$ (*free1* BTH) root sections per genotype, two-tailed unpaired $t$ test; [***]$p < 0.001$. **e** 3D electron tomography (3D-ET) analysis of autophagosomal structures in homozygous *free1* T-DNA insertional mutant upon BTH treatment for 8 h. Left panel, individual slice of the 3D-tomography. Right panel, 3D model of the tomography. Black dash box indicates the selected 3D-ET structures for 3D model. Arrowheads indicate examples of the unsealed autophagosomal structures. Scale bars, 500 nm. The red color indicates the autophagosomal structures, while the blue and green color indicate the membrane-bound organelles. All the imaging analysis was repeated at least for three times with similar results.

accumulated NBR1 and ATG8 can be significantly degraded upon proteinase K digestion in the FREE1 depletion mutants (Supplementary Fig. 5b), supporting that the autophagosomes accumulated in the FREE1 depletion mutants are unclosed. Further analysis showed that more accumulation of both the autophagic-receptor NBR1 and ATG8 upon DEX+BTH+ConcA treatments comparing with the DEX+BTH treatments (Supplementary Fig. 5c). To determine the morphology of autophagosomes in FREE1 mutants, we germinated the heterozygous T-DNA insertional mutants of FREE1 (free1+/-) and selected the lethal seedlings which were the homozygous (free1-/-) mutants[48] for ultrastructural transmission electron microscopy (TEM) and 3D electron tomography (ET) analysis. TEM analysis using the homozygous free1 T-DNA insertional mutants showed the accumulation of numerous unclosed phagophores in the cytoplasm upon BTH treatment, in sharp contrast to the wild type (WT) cells (Fig. 1c, d and Supplementary Fig. 2d). Immunogold-TEM analysis proved that these unclosed structures in free1 mutants were positive for the autophagic-markers including ATG1, ATG8, and ATG13 (Supplementary Fig. 6). Further immunogold-TEM analysis on the transgenic plants expressing YFP-ATG8e in free1 mutants showed that GFP antibodies can substantially label the unclosed autophagosomal structures (Supplementary Fig. 7a), while the GFP antibodies labelling on other intracellular compartments or aggregates could be barely observed (Supplementary Fig. 7b), which are consistent with the live-cell imaging analysis showing the unclosed autophagosmal structures labeled by YFP-ATG8e in FREE1 depletion mutants (Fig. 1a, b, Supplementary Fig. 3, Supplementary Movie. 1, and Supplementary Movie. 2). To further consolidate that the autophagosomes are unclosed in nature, we performed the serial-2D TEM analysis to differentiate the initiated phagophore and unsealed autophagosome, and showed that most of the autophagosomes accumulated in free1 mutants are unsealed, although they seem to be initiated phagophores or closed autophagosomes in some sections (Supplementary Fig. 8). 3D-tomograms further demonstrated the substantial accumulation of unsealed autophagosomes in free1 mutant (Fig. 1e, Supplementary Fig. 9, and Supplementary Movie. 3). Taken together, these results indicate that FREE1 is required for autophagosome closure.

## ATG conjugation system and its substrates ATG8 coordinate with FREE1 for autophagosome closure

To further elucidate the mechanism underlying FREE1 function in autophagosome closure, we performed multiple rounds of affinity purification-mass spectrometry (AP-MS) analysis using FREE1 as the bait upon carbon starvation (Supplementary Data 1). The autophagosome closure is a relatively transient cellular process, making it challenging to identify its regulators until now. We performed analysis on the MS data through a relative quantification by the IBAQ using MaxQuant software[55]. Although a number of proteins have been identified to be enriched in the GFP-FREE1 groups, we have been focusing on candidates function in MVB sorting/autophagy pathways in the current study because FREE1 performs multiple functions in plants and our study focuses on its function in autophagosome closure. Our analysis showed that the FREE1-interacting candidates that function in MVB sorting/autophagy pathways were enriched in the GFP-FREE1 groups compared to the GFP controls as indicated by the IBAQ (Supplementary Data 1 and Supplementary Fig. 10). Under nutrient starvation, FREE1 associated with ESCRTs, HOPS (homotypic fusion and vacuole protein sorting), Retromer, and TOR kinase complex (Fig. 2a). Interestingly, ATG5 was amongst the interactors, suggesting that the ATG conjugation system may recruit FREE1 to the phagophore for closure (Fig. 2a). Although the yeast two hybrid and FRET analysis showed that FREE1 did not exhibit a direct interaction with components of the ATG conjugation system ATG5, ATG12, ATG16 or other ATG proteins (Supplementary Fig. 11), colocalization and co-immunoprecipitation (co-IP) experiments in Arabidopsis

protoplasts have shown that the endogenous ATG5, ATG12b and ATG16 associated with FREE1 (Fig. 2b, c). To find out if the ATG conjugation machinery bridged FREE1 to mediate autophagosome closure, we analyzed autophagosome morphology in atg5-1 mutant. 2D-TEM and 3D-electron tomography analysis of autophagosomes in atg5-1 mutant have shown the unsealed autophagosomal structures during autophagy induction, comparing to WT Col-0 cells (Fig. 2d–f, Supplementary Fig. 12, and Supplementary Movie. 4). WIPI-1, the close homolog of ATG18 in plants, has been used to analyze the autophagosomal structures in the ATG5 mutants in mammal[56]. We have thus performed confocal analysis with image deconvolution on the ATG18a-GFP in atg5-1 mutants. In sharp contrast to the Col-0 cells, our results showed that the ATG18a-GFP positive structures are largely unclosed in atg5-1 mutant cells (Supplementary Fig. 13). Surprisingly, we found that FREE1 directly interacted with the substrates of the ATG conjugation system, i.e. the ubiquitin-like proteins ATG8a, ATG8g, and ATG8i (Fig. 3a and Supplementary Fig. 14a). Notably, phylogenetic trees analysis on the ATG8s in different plant species showed that ATG8a, ATG8g, and ATG8i locate in distinct clades of plant ATG8 paralogs (Fig. 3b). Confocal and super-resolution imaging analysis showed that FREE1 colocalized with ATG8a and ATG8i at the phagophore labeled by ATG5 (Fig. 3c, d). Co-IP (co-immunoprecipitation) and FRET (fluorescence resonance energy transfer) analyses further confirmed the interaction between FREE1 and the specific ATG8 isoforms of ATG8a and ATG8i (Fig. 3e, f). Western blot and quantification analysis further showed the accumulation of full-length eYFP-ATG8i in atg5-1 mutants as well as full-length mCherry-ATG8i in atg5-1 mutants compared to WT Col-0 cells upon autophagic induction (Supplementary Fig. 14b, c). Coherently, the ATG conjugation system components accumulated in the FREE1 depletion mutants (Supplementary Fig. 15). In summary, we demonstrated that the ATG conjugation system and its specific substrates may coordinate with FREE1 on the phagophore for autophagosome closure.

## FREE1 contains a classical AIM motif essential for the interaction with ATG8

Using Yeast-two hybrid screening, Marshall et al. has shown that ATG8 may interact with FREE1 via typical LDS (LIR docking site)[57], suggesting that FREE1 may contain the classical AIM motif. Our in silico analysis found that W438 I441 on FREE1 could be the potential AIM motif essential for its interaction with ATG8 (Supplementary Fig. 16a). We have thus performed mutagenesis on FREE1 by substituting W438 I441 to A438 A441 to investigate whether FREE1 interacts with ATG8 through the classical AIM motif. Indeed, co-immunoprecipitation (co-IP) and yeast two hybrid analysis showed that mutation of W438 I441 to A438 A441 abolished the interaction between FREE1 and ATG8 (Supplementary Fig. 16b, c). Notably, mutation of the AIM motif on FREE1 significantly inhibited the recruitment of FREE1 to the nascent autophagosomes comparing with wild-type FREE1 in Arabidopsis protoplasts (Supplementary Fig. 16d, e), demonstrating that this AIM-dependent interaction was important for the recruitment of FREE1 to the nascent autophagosomes. To further elaborate the function of FREE1 AIM in autophagosome closure, we used the particle bombardment to transfer and express the FREE1 or FREE1W438AI441A in the eYFP-ATG8e/DEX::RNAi-FREE1 transgenic plants for confocal observation of the autophagosomal structures. Indeed, the mature ring-like autophagosomes can be found in the eYFP-ATG8e/DEX::RNAi-FREE1 transgenic plants upon FREE1 overexpression, while the FREE1W438AI441A overexpression cannot complement the defective of autophagosome closure in eYFP-ATG8e/DEX::RNAi-FREE1 (Supplementary Fig. 16f). Taken together, our results demonstrated that FREE1 contains a classical AIM motif, which is essential for its interaction with ATG8, recruitment to autophagosome, and function in autophagosome closure.

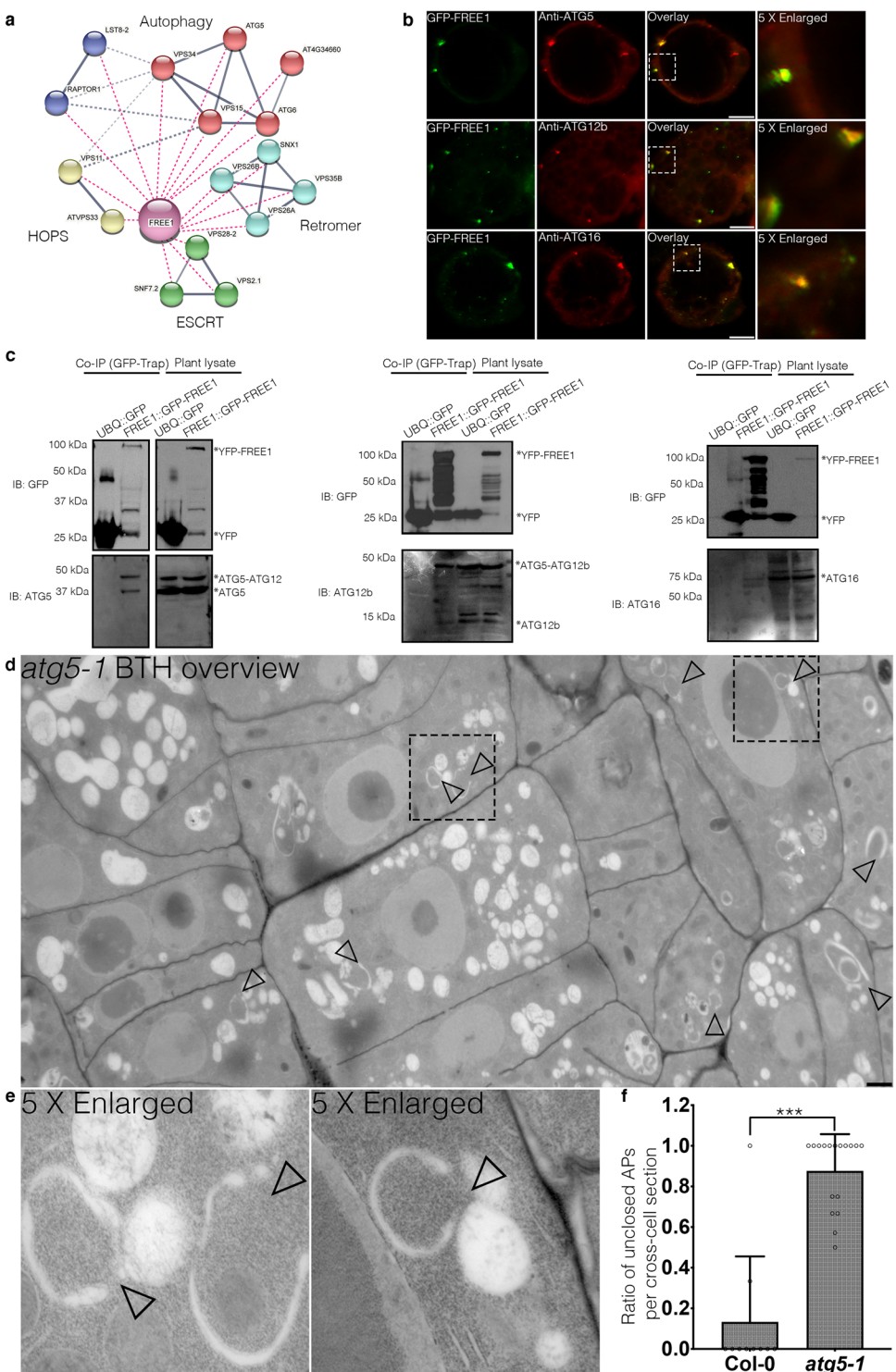

**Fig. 2 | FREE1 interacts with ATG conjugation system and ESCRT-III under starvation conditions. a** Transgenic *Arabidopsis* seedlings expressing YFP-FREE1 upon autophagy induction by carbon starvation for at least 18 h were subjected to GFP-Trap assay for IP-MS analysis. Identified FREE1-interacting partners were analyzed using STRING. **b** Immunofluorescence labeling of the endogenous ATG conjugation system proteins ATG5, ATG12b or ATG16 in *Arabidopsis* protoplasts transfected with GFP-FREE1 for 24 h before confocal laser scanning microscopy (CLSM) observation. White dash boxes indicate the 5 X enlarged areas. Scale bars, 10 μm. **c** GFP-Trap and co-IP analysis of GFP-FREE1 with endogenous ATG5, ATG12b, or ATG16 using native promoter driven GFP-FREE1 transgenic plants. *Arabidopsis* transgenic plants expressing GFP or GFP-FREE1 driven by the native promoter were

subjected to protein extraction and IP with GFP-Trap, followed by western blot analysis with indicated antibodies. **d** Structural TEM analysis of high-pressure freezing/frozen substituted *atg5-1* mutant plant root tips upon BTH treatment for 8 h. Black dash boxes indicated the 5 X zoom-in areas. Arrowheads indicated examples of the unsealed autophagosomal structures. Scale bars, 500 nm. **e** 5 X zoom-in areas from **d**. Arrowheads indicated examples of the unsealed autophagosomal structures. Scale bars, 500 nm. **f** Quantification analysis of the ratio of the unclosed autophagosomal structures (APs) in Col-0 and *atg5-1* mutants per cross-cell section. Means±SD; $n = 10$ (Col-0) and $n = 17$ (*atg5-1*) cells, two-tailed unpaired $t$ test; ***$p < 0.001$. All the imaging analysis and immunoblots were repeated at least for three times with similar results.

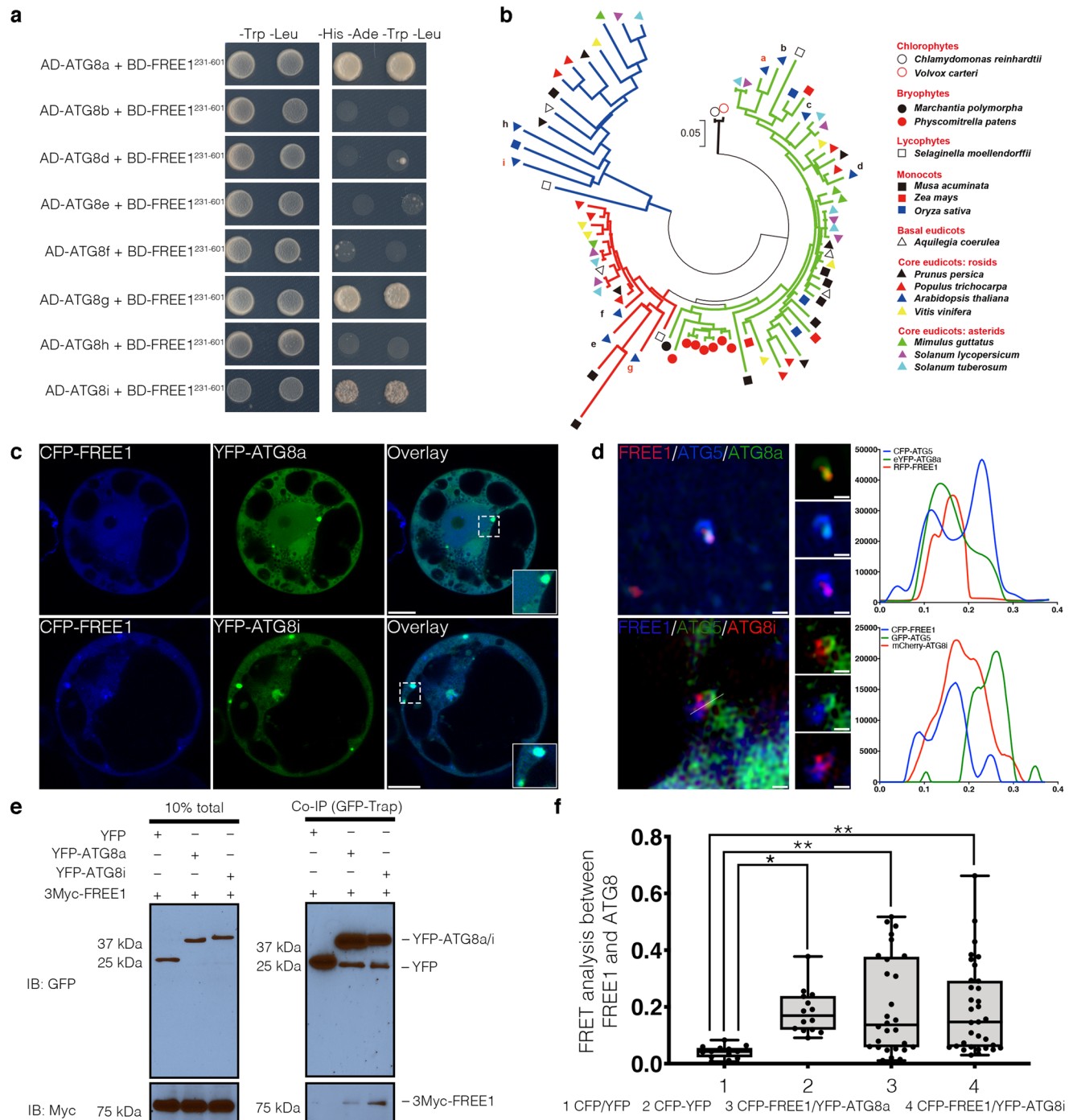

**Fig. 3 | FREE1 specifically interacts with the distinct paralogs of ATG8. a** Y2H analysis of BD-FREE1$^{231–601}$ (the truncated FREE1 without self-activation on BD vector) with AD-ATG8 isoforms. **b** Evolutionary relationships of ATG8 isoforms. The optimal tree with the sum of branch length=4.32642842 is shown. The tree is drawn to scale, with branch lengths in the same units as those of the evolutionary distances used to infer the phylogenetic tree. **c** Transient expression of CFP-FREE1 with specific ATG8 isoforms YFP-ATG8a and YFP-ATG8i in *Arabidopsis* protoplasts, the transfected protoplasts were cultured for 24 h before CLSM observation. White dash boxes indicate the 2.5 X enlarged insets. Scale bars, 10 μm. **d** *Arabidopsis* protoplasts transfected with FREE1, ATG5 and ATG8a or ATG8i were subjected to structure illuminated microscopy (SIM) analysis. Scale bars, 1 μm. **e** GFP-Trap and co-immunoprecipitation (co-IP) analysis of 3Myc-FREE1 with YFP-ATG8a or YFP-ATG8i using *Arabidopsis* protoplasts. *Arabidopsis* protoplasts co-expressing 3Myc-FREE1 with YFP-ATG8a, YFP-ATG8i, or YFP only were subjected to protein

extraction and co-IP with GFP-Trap, followed by immunoblotting with indicated antibodies. **f** FRET analysis between CFP-FREE1 and YFP-ATG8a or YFP-ATG8i in *Arabidopsis* protoplasts. *Arabidopsis* protoplasts were transfected with CFP and YFP (negative control), CFP-YFP (positive control), CFP-FREE1 and YFP-ATG8a, or CFP-FREE1 and YFP-ATG8i, followed by cultured for 24 h before FRET analysis. Means±SD; $n = 14$ (CFP/YFP), $n = 14$ (CFP-YFP), $n = 28$ (CFP-FREE1/YFP-ATG8a), and $n = 33$ (CFP-FREE1/YFP-ATG8i) individual puncta per experimental group, one-way analysis of variance (ANOVA), followed by Dunnett's multiple comparisons test; $^*p < 0.05$; $^{**}p < 0.01$. The middle lines of the boxes represent the medians of datasets. The upper and bottom lines of the boxes are the upper quantile and the lower quantile of the data, respectively. The whiskers mark the upper and lower limits of these datasets, respectively. All the imaging analysis and immunoblots were repeated at least for three times with similar results.

### ESCRT-III components are recruited by FREE1 onto the autophagosomes to mediate closure

In mammal, ESCRTs components VPS37 and SNF7 are essential for autophagosome closure[31–33]. In yeast, Rab5 GTPase seems to control the recruitment of ESCRT to unsealed autophagosomes through an interaction of the ESCRT subunit SNF7 with Atg17[34]. Several ESCRT components are identified as FREE1 interactors upon starvation (Fig. 2a), we thus set out to investigate the role of ESCRT-III complex in autophagosome closure in plants. To illustrate the ESCRT-III function in plant autophagosome closure, we generated transgenic plants co-expressing eYFP-ATG8e and SNF7.1-RFP, a plant ESCRT-III component[58]. Under nutrient rich conditions, SNF7.1 mainly exhibited cytosolic and punctate patterns, while ATG8e was mainly localized in the cytoplasm (Supplementary Fig. 17a). Upon autophagic induction by BTH treatment, ATG8e-positive punctae appeared and significantly colocalized with SNF7.1 (Supplementary Fig. 17a, b). Further analysis using transient expression in *Arabidopsis* protoplasts demonstrated that SNF7.1 not only exhibited similar localization pattern as previously shown[59], but also localized at the phagophore (Supplementary Fig. 17c). Consistently, another ESCRT-III component VPS20 also localized to the autophagosomes upon autophagic induction (Supplementary Fig. 18a). To analyse the function of ESCRT-III in autophagosome closure, we then generated dominant-negative mutant plants expressing SNF7.1(L22W) (SNF7.1DN), which has been shown to cause the enlargement of the PVC/MVB with a reduced number of intraluminal vesicles (ILVs)[59]. Live cell imaging showed that unsealed ATG8e-positive autophagosomes accumulated substantially while the formation of autophagic bodies were significantly impaired in the SNF7.1DN mutants (Supplementary Fig. 17d, e). Notably, TEM analysis of autophagosomal structures in SNF7.1DN demonstrated that the accumulated autophagosomes in the cytoplasm were unsealed (Supplementary Fig. 17f). 3D electron tomography analysis further confirmed the unsealed nature of autophagosomes in SNF7.1DN mutants (Supplementary Fig. 17g and Supplementary Movie. 5). Consistently, dominant negative mutants of VPS4(E232Q) (VPS4DN), the ATPase-deficient form of VPS4 that causes the enlargement of the PVC/MVB with a reduced number of ILVs[38], also accumulated unsealed ATG8e labelled autophagosomes (Supplementary Fig. 18b, c) and showed significant defects in vacuolar turnover autophagic flux (Supplementary Fig. 18d, e). We next investigated whether FREE1 is required for the recruitment of the ESCRT-III components onto autophagosomal structures to regulate closure. Immunogold-TEM analysis using the antibodies against endogenous plant ESCRT-III component SNF7[58] showed that SNF7 can significantly label the autophagosomal structures in Col-0 wild-type plants, but not the accumulated unclosed autophagosomal structures in *free1* mutants (Supplementary Fig. 19a, b). Our results thus proved that FREE1 is important for the recruitment of other ESCRT components including ESCRT-III onto the autophagosomes. To further consolidate the TEM observations, we next performed the transient expression assay using protoplasts isolated from *Arabidopsis* PSBD suspension culture cells for confocal analysis. Consistently, SNF7.1-GFP failed to be recruited onto the ATG8-positive structures labelled by mCherry-ATG8e in the protoplasts expressing the FREE1 RNAi (Supplementary Fig. 19c, d). Notably, SNF7.1-GFP failed to be recruited onto the ATG8-positive structures in the protoplasts expressing FREE1 RNAi and the AIM motif mutant Myc-FREE1^W438AI441A (Supplementary Fig. 19c, d). Nonetheless, the ESCRT-III component SNF7.1-GFP could be sufficiently recruited onto the ATG8-positive structures in the protoplasts expressing FREE1 RNAi and wild-type Myc-FREE1 (Supplementary Fig. 19c, d). Western blot analysis showed the proper overexpression of the FREE1 protein and its mutant variants in the FREE1 RNAi mutants (Supplementary Fig. 19e). Therefore, our results demonstrated that the interaction between FREE1 and ATG8 is required for the recruitment of the ESCRT-III components onto the autophagosome.

Recent studies have demonstrated the importance of ESCRT-I in regulating autophagosome closure in mammal[32], we thus also set out to investigate the potential function of plant ESCRT-I in phagophore sealing. In *Arabidopsis*, all the ESCRT-I components VPS37, VPS23, and VPS28 contain two homologues encoding in the genome. Double mutant of the ESCRT-I components is inapplicable for cellular analysis due to the seedling lethality, while the single mutant does not exhibit obvious growth defective under normal condition[60]. Consistent with recent study[54], phenotypic analysis showed that neither the single mutant *vps23a* nor *vps23b* mutants exhibited comparable autophagy-related phenotype as *atg5-1* and *atg7-2* upon carbon starvation (Supplementary Fig. 20a). TEM results showed that the autophagosomes closed normally in *vps23a-/-vps23b + /-* mutant, which is comparable with the wild-type plants (Supplementary Fig. 20b). Therefore, our current results suggested that VPS23A may either act redundantly with VPS23B in autophagosome closure or do not play role in autophagy. Due to the lethality of the double mutants of VPS23, future study could be carried out to generate micro-RNA mutants of both VPS23A and VPS23B driven by inducible promoter and used for exploring the possible functional roles of ESCRT-I in autophagosome closure. Taken together, our results demonstrated that the interaction between FREE1 and ATG8 is required for the recruitment of the ESCRT-III components onto the autophagosomal structures to regulate autophagosome closure in plants.

### Plant energy sensor kinase SnRK1 phosphorylates FREE1

AMPK is a conserved metabolic switch that senses cellular energy status and governs energy homeostasis through its regulation of glucose and lipid metabolism in mammal. The regulatory roles of the AMPK on autophagosome initiation through phosphorylation and activation of ULK1[6–8] and differential regulation of VPS34 containing complexes[9] have been unveiled. In plants, SnRK1 (Sucrose non-fermenting-1 (SNF1)-related kinase 1), the close homolog of the yeast SNF1 (sucrose non-fermenting 1) and the mammalian AMPK families, is a central integrator of energy and stress signalling. Upon sensing energy deficit, nutrient deprivation or darkness, SnRK1 triggers vast metabolic and transcriptional adjustments for homeostasis restoration, survival promotion and environmental adaptation[61]. Surprisingly, our interactome analyses revealed that FREE1 associates with KIN10 (SnRK1α1) during nutrient starvation (Supplementary Data 1 and Supplementary Fig. 10). We thus investigated the localization of FREE1 with KIN10 in *Arabidopsis* protoplasts. Interestingly, FREE1 partially colocalized with KIN10 (Fig. 4a). Both FRET and yeast-two hybrid assay demonstrated the direct interaction between FREE1 and KIN10 (Fig. 4b, c), while co-IP assays in *Arabidopsis* protoplasts further showed that GFP-FREE1 can be co-immunoprecipitated with KIN10-6HA (Fig. 4d). Finally, the in vivo analysis using KIN10-GFP transgenic lines has demonstrated that KIN10 interacts with FREE1 in a nutrient starvation dependent manner (Fig. 4e, f and Supplementary Fig. 21a). In vivo interaction assay using the endogenous promoter driven GFP-FREE1 transgenic plants further confirmed that FREE1 can directly interact with the endogenous KIN10 upon nitrogen starvation (Supplementary Fig. 21b). The interaction between KIN10 and FREE1 prompted us to hypothesize that KIN10 might phosphorylate FREE1 to promote autophagosome closure during nutrient starvation. Indeed, FREE1 can be phosphorylated under nitrogen limiting conditions comparing with mock treatments (Fig. 4g). To identify the potential phosphorylation sites on FREE1, we performed multiple rounds of pull-down and mass-spectrometry analysis during nutrient starvation. Interestingly, Serine at position 530 of FREE1, was phosphorylated upon carbon deprivation (Supplementary Fig. 22a, b). To investigate whether KIN10 can directly phosphorylate FREE1, we performed the semi-in vitro phosphorylation assay. According to previous study[62], KIN10 was first enriched from transgenic seedlings overexpressing KIN10-GFP or KIN10-HA by immunoprecipitation with anti-GFP or anti-HA and then used to

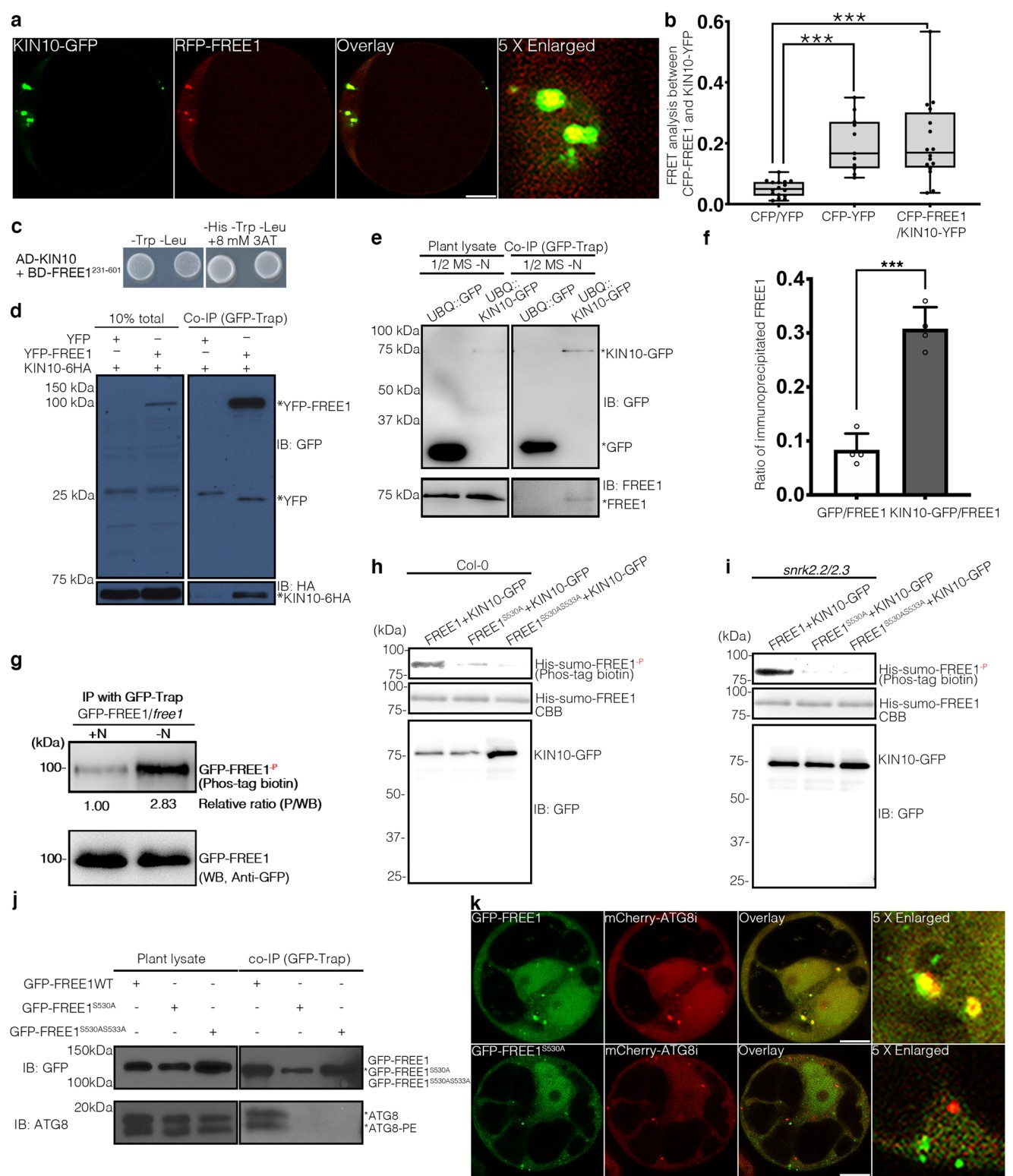

phosphorylate His-sumo-FREE1, His-sumo-FREE1^S530A, or His-sumo-FREE1^S530AS533A since it also contained the S530A mutation, which was further detected by Phos-tag biotin. Indeed, FREE1 can be phosphorylated in the presence of the activated KIN10-GFP or KIN10-HA, while the negative control GFP only cannot be phosphorylated (Fig. 4h and Supplementary Fig. 23a, b). Importantly, mutation on S530A abolished the phosphorylation of FREE1 by KIN10 (Fig. 4h). FREE1 has been shown to be phosphorylated by SnRK2.2/2.3, promoting its translocation to the nuclei under ABA conditions[63], whereas a recent

study has also shown the sequestration of SnRK1 by SnRK2-containing complexes to inhibit SnRK1 signaling[64]. Therefore, to provide the direct evidence of FREE1 phosphorylation by SnRK1 but not by SnRK2.2/2.3 in the semi-in vivo phosphorylation assay, we also used the Arabidopsis protoplasts isolated from the snrk2.2/2.3 mutants for the assay. Consistently, FREE1 can be phosphorylated in the presence of KIN10 purified from the transgenic seedlings expressing KIN10-GFP in snrk2.2/2.3 mutants background, while the mutation on S530A or S530AS533A abolished the phosphorylation of FREE1 by SnRK1α1

**Fig. 4 | KIN10 phosphorylates FREE1 to orchestrate ESCRT machinery function in autophagosome closure. a** Transient expression and confocal analysis of the spatial relationship between FREE1 and KIN10 in *Arabidopsis* leaf protoplasts. The transfected leaf protoplasts were cultured for 24 h before CLSM observation. Scale bars, 10 μm. **b** FRET analysis of CFP-FREE1 with KIN10-YFP. *Arabidopsis* protoplasts co-expressing CFP and YFP (negative control), CFP-YFP (positive control), or CFP-FREE1 and KIN10-YFP were cultured for 24hr before FRET analysis. Means±SD; $n = 16$ (CFP/YFP), $n = 11$ (CFP-YFP), and $n = 16$ (CFP-FREE1/KIN10-YFP) individual puncta per experimental group, one-way analysis of variance (ANOVA), followed by Tukey's multiple test; ***$p < 0.001$. The middle lines of the boxes represent the medians of datasets. The upper and bottom lines of the boxes are respectively the upper quantile and the lower quantile of the data. The whiskers mark the upper and lower limits of these datasets, respectively. **c** Y2H analysis between AD-KIN10 and BD-FREE1[231-601] (the truncated FREE1 without self-activation on BD vector). **d** GFP-Trap and co-immunoprecipitation (co-IP) analysis of KIN10-6HA and YFP-FREE1 in *Arabidopsis* protoplasts. *Arabidopsis* protoplasts co-expressing KIN10-6HA with YFP-FREE1, or YFP only were subjected to protein extraction and IP with GFP-Trap, followed by immunoblotting with indicated antibodies. **e** GFP-Trap and co-IP of KIN10 and FREE1 using transgenic plants expressing KIN10-GFP under nitrogen starvation (-N) for at least 18hrs, followed by immunoblotting using GFP and FREE1 antibodies. **f** Quantification analysis of the ratio of co-immunoprecipitated FREE1 by KIN10 shown in **e**. Means±SD; $n = 4$ individual experiments, two-tailed unpaired $t$ test, ***$p < 0.001$. **g** In vivo phosphorylation of FREE1 upon nitrogen starvation. GFP-

FREE1/*free1* transgenic plants were subjected to mock or nitrogen starvation for at least 18 h, followed by protein extraction and IP with GFP-Trap, and subsequent immunoblotting detection by Phos-tag biotin and the indicated antibodies. **h** Semi-in vitro phosphorylation assay of FREE1, FREE1[S530A] and FREE1[S530AS533A] proteins by KIN10-GFP purified from the transgenic plants expressing KIN10-GFP. KIN10 was enriched from transgenic seedlings expressing KIN10-GFP by immunoprecipitation with anti-GFP and then used to phosphorylate FREE1, FREE1[S530A] or FREE1[S530AS533A] tagged with His-sumo, which was further detected by Phos-tag biotin. **i** Semi-in vitro phosphorylation assay of FREE1, FREE1[S530A] and FREE1[S530AS533A] proteins by KIN10-GFP purified from the transgenic plants expressing KIN10-GFP in *snrk2.2/2.3* mutants. KIN10 was enriched from transgenic seedlings expressing KIN10-GFP in *snrk2.2/2.3* mutants background by immunoprecipitation with anti-GFP and then used to phosphorylate FREE1 or FREE1[S530A] or FREE1[S530AS533A] tagged with His-sumo, which was further detected by Phos-tag biotin. **j** In vivo interaction between ATG8 and FREE1, FREE1[S530A], or FREE1[S530AS533A] mutant variants. GFP-FREE1, GFP-FREE1[S530A], and GFP-FREE1[S530AS533A] transgenic plants were subjected to nitrogen starvation for at least 18 h, followed by protein extraction and IP with GFP-Trap, and subsequent immunoblotting with indicated antibodies. **k** Transient co-expression of GFP-FREE1 or GFP-FREE1[S530A] with mCherry-ATG8i in *Arabidopsis* protoplasts. The transfected protoplasts were cultured for 24 h before CLSM observation. Scale bars, 10 μm. All the imaging analysis and immunoblots were repeated at least for three times with similar results.

---

(Fig. 4i). Furthermore, the in vitro kinase assays were performed using recombinant KIN10 tagged with His-MBP and FREE1 or FREE1[S530A] tagged with His-sumo, which was further detected by the autoradiography. Indeed, FREE1 can be phosphorylated by the purified KIN10, while the mutation on S530A abolished the phosphorylation of FREE1 by KIN10 in vitro (Supplementary Fig. 23c, d). To further confirm our results *in planta*, GFP-FREE1 or GFP-FREE1[S530A] were transiently expressed in protoplasts isolated from transgenic plants expressing KIN10-HA upon nitrogen starvation. GFP-FREE1 and GFP-FREE1[S530A] were enriched by immunoprecipitation with anti-GFP for detection by Phos-tag biotin. Indeed, GFP-FREE1 can be phosphorylated in the protoplasts isolated from the transgenic seedlings expressing KIN10-HA, while the mutation on S530A significantly reduced the phosphorylation of FREE1 by KIN10 (Supplementary Fig. 23e). Taken together, our data demonstrated that KIN10 can directly phosphorylate FREE1 on the S530.

## Phosphorylated-FREE1 is essential for the autophagosome sealing and plant growth upon nutrient deprivation

To investigate the functional role of the phosphorylation site on FREE1 in autophagosome closure, we performed biochemical, genetic and cellular analysis to find out whether the Serine at position 530 is essential for the FREE1 function in autophagosome sealing. The homozygous lines of *free1* T-DNA insertional mutants are seedling lethal, which are not suitable for studying the bona fide function of FREE1 in different developmental stages and growth conditions[48]. Thus, we have screened for the CRISPR mutants of FREE1 which are viable under normal growth conditions and identified the *free1*-ct mutant lines (a *free1* weak allele which harbours a deletion of eight nucleotides in the ninth exon region that leads to a change in amino acids or deletion in the C terminus of FREE1)[63]. Albeit the Ser530 is presented in the protein expressed in the *free1*-ct mutant lines, our recent study suggested that these mutants may disrupt the secondary structure and abolish the function of coiled-coil domain of FREE1 which contains the Ser530[63]. Since Ser530 located at the C-terminus of FREE1 protein, we performed the analysis using this recently obtained CRISPR mutant of FREE1, *free1*-ct[63]. Indeed, albeit showing similar growth phenotype as the wild type plants under normal conditions, *free1*-ct exhibited early senescence phenotypes as *DEX::RNAi-FREE1*, SNF7.1DN, VPS4DN and *atg5* mutants upon cellular energy deprivation and nitrogen starvation (Supplementary Fig. 24a-e). We further explored the functional role of Ser530 of FREE1 in the

autophagy pathway using the transgenic plants expressing GFP-FREE1 or GFP-FREE1[S530A] in *free1*-ct mutant background, as well as GFP-FREE1[S530AS533A]/*free1*-ct that we recently generated since it also contained the S530A mutation[63]. Indeed, *free1*-ct plants exhibited growth defects when cellular energy and nitrogen source were limited, however, overexpressing GFP-FREE1[S530A] or GFP-FREE1[S530AS533A] was unable to complement the phenotype, compared to WT Col-0 and GFP-FREE1 complementation lines (Supplementary Fig. 24f, g and Supplementary Fig. 25a, b). We have also generated the phosphomimetic variants of FREE1 transgenic plants expressing the GFP-FREE1[S530D] or GFP-FREE1[S530DS533D] in *free1*-ct mutants background for phenotypic analysis. Consistently, overexpressing of GFP-FREE1[S530D]/*free1*-ct or GFP-FREE1[S530DS533D]/*free1*-ct was able to complement the defective phenotype as the GFP-FREE1 complementation lines upon carbon starvation (Supplementary Fig. 25c, d). These results suggested that the phosphorylated-FREE1 is indispensable for the plant autophagy.

Notably, GFP-FREE1 can interact with ATG8, while GFP-FREE1[S530A] and GFP-FREE1[S530AS533A] cannot be co-immunoprecipitated with ATG8 in planta (Fig. 4j and Supplementary Fig. 26a). Indeed, in vivo interaction assay using the endogenous promoter driven GFP-FREE1 transgenic plants further confirmed that FREE1 can directly interact with the endogenous ATG8 upon nitrogen starvation (Supplementary Fig. 26b). Cellular analysis showed the decreased colocalization of GFP-FREE1[S530A] with ATG8i in *Arabidopsis* protoplasts (Fig. 4k and Supplementary Fig. 26c), and partial decrement of vacuolar delivery of GFP-FREE1[S530A] or GFP-FREE1[S530AS533A] in comparison to GFP-FREE1, GFP-FREE1[S530D] or GFP-FREE1[S530DS533D] (Supplementary Fig. 27a, b). Intriguingly, GFP-FREE1[S530A] failed to co-immunoprecipitate with the ATG conjugation system as well, while GFP-FREE1 can be significantly precipitated by ATG5 and ATG5-ATG12 conjugate (Supplementary Fig. 28 and Fig. 2c), suggesting the importance of S530 for the FREE1-ATG conjugation system interaction. To further elaborate the role of S530 on FREE1 in autophagosome closure, 2D-TEM and 3D-tomography analysis further showed the accumulation of unclosed autophagosomal structures in the *free1-ct*, GFP-FREE1[S530A]/*free1-ct*, and GFP-FREE1[S530AS533A]/*free1-ct* lines, while the autophagosomes in the GFP-FREE1[S530D]/*free1-ct*, GFP-FREE1[S530DS533D]/*free1-ct*, and GFP-FREE1/*free1-ct* complementation lines were properly sealed upon carbon starvation (Fig. 5a, b and Supplementary Movie. 6). Consistently, we monitored the autophagy flux as previous study[65] and showed that the autophagic bodies were largely absent from the vacuole of the *free1-ct*,

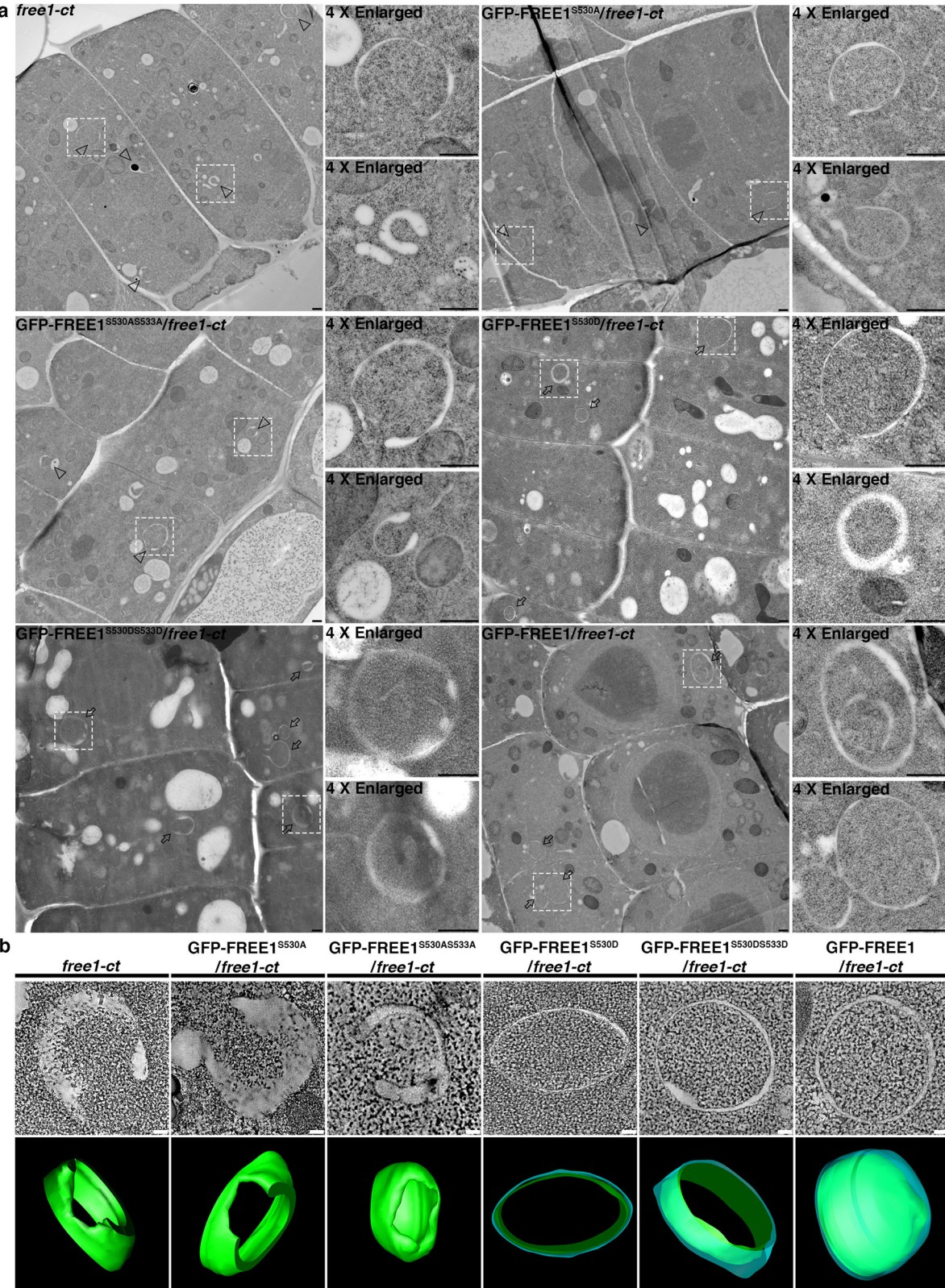

**Fig. 5 | The phosphorylation site on FREE1 is essential for the autophagosome closure. a** Structural TEM analysis of ultrathin sections prepared from high-pressure freezing/frozen substituted *free1-ct* mutant, GFP-FREE1^S530A/*free1-ct*, GFP-FREE1^S530AS533A/*free1-ct*, GFP-FREE1^S530D/*free1*-ct, GFP-FREE1^S530DS533D/*free1-ct*, and GFP-FREE1/*free1-ct* transgenic plant root tips upon carbon starvation for 18 h. White dash boxes indicate the 4 X zoom-in areas. Arrowheads indicate examples of the unsealed autophagosomal structures, while arrows indicate examples of the sealed autophagosomes. Scale bars, 500 nm. **b** 3D electron tomography (ET) analysis of autophagosomal structures in *free1-ct* mutant, GFP-FREE1^S530A/*free1-ct*, GFP-FREE1^S530AS533A/*free1-ct*, GFP-FREE1^S530D/*free1-ct*, GFP-FREE1^S530DS533D/*free1-ct*, and GFP-FREE1/*free1-ct* transgenic plant root tips upon carbon starvation for 18 h. Upper panels, individual slices of the 3D-tomography. Lower panels, 3D models of the corresponding tomography. Scale bars, 100 nm. The green color indicates the unclosed autophagosomal structures, while the blue and green color indicate the closed autophagosomes. All the imaging analysis was repeated at least for three times with similar results.

GFP-FREE1[S530A]/*free1-ct*, and GFP-FREE1[S530AS533A]/*free1-ct* lines, yet the autophagic bodies were substantially accumulated in the vacuoles of the GFP-FREE1[S530D]/*free1-ct*, GFP-FREE1[S530DS533D]/*free1-ct*, and GFP-FREE1/*free1-ct* complementation lines upon carbon starvation and ConcA treatments, suggesting the defective of vacuolar delivery of the unclosed autophagosomal structures in *free1-ct*, GFP-FREE1[S530A]/*free1-ct*, and GFP-FREE1[S530AS533A]/*free1-ct* lines (Supplementary Fig. 29a–c).

Biochemically, the lipidated ATG8 significantly accumulated in the *DEX::RNAi-FREE1* mutant protoplasts overexpressing Myc-FREE1[S530A], *DEX::RNAi-FREE1* mutant protoplasts overexpressing Myc-FREE1[W438AI441A], as well as *DEX::RNAi-FREE1* mutant protoplasts upon DEX induction with or without BTH treatments, while the effects could be restored by overexpressing wild-type Myc-FREE1 in *DEX::RNAi-FREE1* mutant protoplasts (Supplementary Fig. 30a, b). Meanwhile, the autophagy receptor NBR1 substantially accumulated in the *DEX::RNAi-FREE1* mutant protoplasts overexpressing Myc-FREE1[S530A], *DEX::RNAi-FREE1* mutant protoplasts overexpressing Myc-FREE1[W438AI441A], as well as *DEX::RNAi-FREE1* mutant protoplasts upon DEX induction with or without BTH treatments, while the defects could be recovered by expressing wild type Myc-FREE1 in *DEX::RNAi-FREE1* mutant protoplasts (Supplementary Fig. 30c, d). Lastly, the YFP-ATG8e turnover was significantly impeded in the *DEX::RNAi-FREE1* mutant protoplasts overexpressing Myc-FREE1[S530A], *DEX::RNAi-FREE1* mutant protoplasts overexpressing Myc-FREE1[W438AI441A], as well as *DEX::RNAi-FREE1* mutant protoplasts upon DEX induction with or without BTH treatments, while the defects could be complemented by overexpressing wild type Myc-FREE1 in *DEX::RNAi-FREE1* mutant protoplasts (Supplementary Fig. 30e, f). Our results thus demonstrated that FREE1[S530A] failed to complement the defective of FREE1 mutants on ATG8 lipidation, NBR1 turnover, and YFP-ATG8e turnover, indicating that S530 is essential for FREE1 function in autophagosome closure biochemically. Albeit the importance in regulating autophagosome closure, cellular and biochemical analysis showed that S530 on FREE1 is dispensable for the vacuolar transport of Aleurain-mRFP (Supplementary Fig. 31). Our results thus demonstrated that the S530 affects only autophagy but not vacuolar protein transport. To conclude, our data demonstrated that the phosphorylation status of FREE1 is essential for its redistribution to phagophore, function in autophagosome closure, and dedication to plant growth upon nutrient deprivation.

## Discussion

Using a combination of 3D electron tomography, proteomic analysis, cellular and genetic approaches, we demonstrated that the ATG conjugation system substrate ATG8 is essential to recruit the ESCRTs machinery for autophagosome completion via a plant unique pleiotropic ESCRT protein FREE1, whose shuttling from endosome to phagophore is tightly regulated by the plant energy sensor SnRK1, by which ATG8 interacts and recruits SnRK1α1-phosphorylated FREE1 onto phagophores for downstream ESCRTs recruitment to regulate autophagosome closure. Our study thus provides hints on the universal underlying mechanism of nutrient sensing and autophagosome closure.

Despite the indispensable role of the ESCRT machinery in autophagosome closure[31–33], exactly how the ESCRT components are translocated from endosome to phagophore upon autophagy induction by nutrient starvation remains unknown in higher eukaryotes. Consistently our study confirmed that the plant ESCRT-III proteins SNF7.1 and VPS20 localize at the autophagosomal structures, while unclosed autophagosomes largely accumulated in SNF7.1 and VPS4 mutants (Supplementary Figs. 17, 18). More intriguingly, we demonstrated that the plant unique ESCRT component FREE1, which has been shown to regulate endosomal sorting and MVB/Vacuole biogenesis in concert with ESCRT-I in *Arabidopsis*[48], is crucial for autophagosome closure. Substantial numbers of "open" phagophores accumulate in *free1* mutants as revealed by high-resolution confocal imaging, ultra-structural transmission electron microscopy, 3D electron tomography,

and biochemical analysis (Fig. 1a–e, Supplementary Fig. 2, Supplementary Fig. 3, Supplementary Fig. 4, Supplementary Fig. 5, Supplementary Fig. 6, Supplementary Fig. 7, Supplementary Fig. 8, and Supplementary Fig. 9). These results, together with our previous findings that the autophagic bodies could be barely observed in the vacuole of *free1* mutants comparing with the wild-type Col-0 upon BTH and ConcA treatments[49], suggested that the loss of FREE1 in *free1* mutant strongly blocked the autophagosome closure and the unclosed autophagosomes in *free1* mutants could not be properly fused with the vacuole for degradation under autophagy induction conditions. Surprisingly, our proteomic analysis identified the ATG conjugation system as the potential interactors of FREE1 upon nutrient deprivation (Fig. 2a, Supplementary Fig. 10, and Supplementary Data 1). FREE1 colocalizes and can be co-immunoprecipitated with the components of ATG conjugation system (Fig. 2b, c), whereas FREE1 can directly interact with specific paralogs of ATG8, the substrate of the ATG conjugation system (Fig. 3a–f). It would be of great interest in future study to reveal the mechanism underlying the recognition of FREE1 by specific ATG8s in regulating autophagosome closure and to answer whether the mechanism is evolutionary conserved in other plants. Recent studies have shown that depletion of ATG8 proteins apparently causes the accumulation of "open" autophagosomes in mammalian cells[66,67], whereas deficiency in the ATG conjugation system leads to a significant delay in degradation of the inner autophagosomal membrane and cargo, probably due to the incomplete closure of the autophagosomal structures[68]. However, how the ATG conjugation system contributes to autophagosome closure remains elusive, although it has been speculated that ATG8 proteins may be involved in the recruitment of specific LC3-interacting region (LIR)-containing proteins required for downstream phagophore closure[69]. Intriguingly, FREE1 contains a classical AIM motif, which is required for its interaction with ATG8 as well as its recruitment to autophagosome for closure function (Supplementary Fig. 16a–f). Our study has identified and demonstrated that the plant unique ESCRT protein FREE1 acts as a linker to bridge the ATG conjugation system/ATG8s and ESCRT machinery to regulate the autophagosome closure, thus delineating the underlying mechanism of autophagosome sealing in higher eukaryotes.

Plants respond to abiotic and biotic stresses by fine tuning anabolic and catabolic reactions[70]. SnRK1, a central energy sensor kinase and a chief stress response component in plants, is crucial for catabolic reactions and represses anabolic processes upon energy-depleting stress conditions to support stress tolerance and survival[61,71,72]. SnRK1 is functionally and evolutionarily conserved with the SNF1 in yeast and the AMPK in animals[71]. In plants, SnRK1 has been shown to participate in the autophagy pathway, the important catabolic process for energy and nutrient restoration, via acting upstream of TOR or independently of TOR through direct interaction with ATG1[1,73–75]. Here, we provide multiple evidences on KIN10 in regulating the plant unique ESCRT component FREE1 to function in autophagosome closure. We showed that FREE1 and KIN10 directly interacts with each other (Fig. 4b–f and Supplementary Fig. 21). Indeed, FREE1 can be directly phosphorylated by KIN10 in vivo and in vitro, while the phosphorylation sites on FREE1 are essential for the interaction with ATG8 (Fig. 4g–j, Supplementary Fig. 23, and Supplementary Fig. 26). Genetic, cellular and biochemical analysis further supports the notion that KIN10-mediated phosphorylation of FREE1 is essential for autophagosome closure (Fig. 5a, b, Supplementary Fig. 24, Supplementary Fig. 25, Supplementary Fig. 27, Supplementary Fig. 29, and Supplementary Fig. 30). Notably, recent study has shown that the autophagosome formation is strongly inhibited in *kin10* mutant upon a wide variety of stress conditions[75]. Future study using the micro-RNA mutants of SnRK1α1 and SnRK1α1/SnRK1α2 driven by the DEX-inducible promoter[48] could further increase our understanding of the functional roles of SnRK1 in

autophagosome closure. In mammal, AMPK acts on triggering the autophagosome biogenesis through the activation of ULK1[6–8] and regulation of VPS34 containing complexes[9], which are known regulators in autophagy initiation and progression, respectively. Similarly, KIN10 can phosphorylate ATG1 and ATG6 to regulate plant autophagy pathway[62,73]. Our results provided the first line of evidence to show that the energy sensor also plays pivotal role in regulating the downstream closure step of autophagosome biogenesis. The simultaneously targeting of multi-steps during autophagosome formation by KIN10 may ensure the precise regulation of autophagy upon stresses.

Altogether, our results show that during nutrient starvation, the energy sensor kinase not only suppresses the TOR signalling pathway and induce autophagosome initiation, but also regulates the shuttling of ESCRT machinery from endosomes to autophagosomes to facilitate autophagosome sealing (Fig. 6).

## Methods

### Plasmid construction
Restriction digestion or Gateway cloning (Invitrogen) were performed to clone the PCR-amplified target genes into the destination plasmids. For the generation of YFP-ATG8i, mCherry-ATG8i, DEX::SNF7.1(L32W), DEX::VPS4(E232Q), VPS20a-GFP, SNF7.1-RFP, KIN10-GFP, and KIN10-HA transgenic plants, PCR-products of ATG8i, SNF7.1(L32W), VPS4(E232Q), VPS20a, and SNF7.1, KIN10 were cloned into corresponding pBI121/1300-UBQ::Gene-GFP/mRFP, pCambia1300-GFP/mRFP, or pTA7002-35S::Gene dexamethasone (DEX)-inducible backbones by restriction digestion and ligation or Gateway. For transient expression constructs used in confocal laser scanning microscopy (CLSM) or co-IP assays, PCR products were cloned into pBI221-UBQ::XFP-gene, 6HA-gene, or 3Myc-gene vectors by restriction digestion and ligation or Gateway. All constructs were confirmed by Sanger sequencing. Constructs and primers used in this study are listed in Supplementary Data 2.

### Plant growth and phenotypic analysis
*Arabidopsis* plant system biology dark-type culture (PSBD) suspension cells were incubated at 25 °C and sub-cultured every 5 days[76]. *Arabidopsis* seeds were surface sterilized and grown on plates with half-strength Murashige and Skoog (MS) growth medium (pH 5.7). Seeds were germinated and grown at 22 °C under a long day (16 h light/8 h dark) photoperiod. For DEX, BTH, or Concanamycin A (ConcA) treatments in CLSM observation, protein extraction analysis, and high-pressure freezing/frozen, 10 μM DEX, 100 μM BTH, and 1 μM ConcA were applied to seedlings for indicated time points before analysis, respectively. For nitrogen or carbon starvation assay, 5-day-old seedlings were transferred to nitrogen source-free or carbon source-free half-strength MS medium for at least 2 weeks before phenotypic analysis and chlorophyll content measurement.

### Transgenic lines generation
Transgenic plants were generated by introducing the corresponding genes into the target lines through *Agrobacterium* transformation by floral dip[77]. In general, YFP-ATG8i, mCherry-ATG8i, DEX::SNF7.1(L32W), DEX::VPS4(E232Q), VPS20a-GFP, SNF7.1-RFP, KIN10-GFP, and KIN10-HA cloned into the corresponding pBI121/1300-UBQ::Gene-GFP/mRFP, pCambia1300-GFP/mRFP, or pTA7002-35S::Gene dexamethasone (DEX)-inducible backbones were transformed into *Agrobacterium tumefaciens* GV3101 and then introduced into Col-0 or mutant *Arabidopsis* plants via floral dip. Transgenic plants were selected on MS medium plates containing antibiotics, and homozygous lines were used for further analysis. Double transgenic plants were generated by crossing with previously generated marker lines[78].

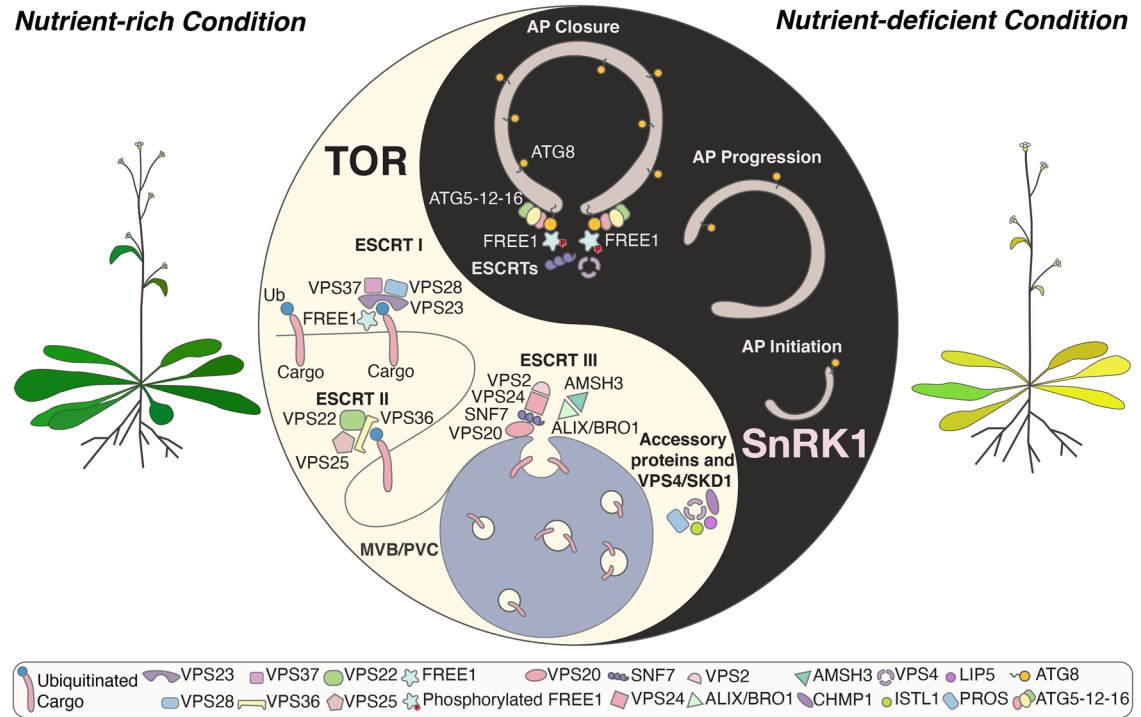

**Fig. 6 | Working model of the interplay between the ATG conjugation system and the ESCRT machinery in regulating autophagosome closure in plants.** Under nutrient-rich condition, FREE1 functions in endosomal sorting. During nutrient-deficient condition (autophagy induction), SnRK1 phosphorylates FREE1. The phosphorylated FREE1 is then recruited by the ATG conjugation system/ATG8 isoforms and cooperates with the downstream ESCRT machinery to mediate autophagosome closure. TOR, target of rapamycin; SnRK1, SNF1-related protein kinase 1; AP, autophagosome; MVB/PVC, multivesicular body/prevacuolar compartment; ESCRT, endosomal complexes required for transport.

## Confocal microscopy and FRET analysis

Confocal fluorescence images were captured by a Leica SP8 laser scanning confocal system with a 63 x water lens and deconvolution. For each analysis, >30 individual cells or >10 individual seedlings were analyzed to represent the majority of samples showing similar expression levels and patterns. Images were further processed by Adobe Photoshop software (http://www.adobe.com). Deconvolution was performed via HyVolution software from Leica. FRET analysis was conducted on the same Leica SP8 confocal system according to the manufacturer's instruction using 405 nm and 514 nm laser. CFP -YFP indicates the fusion of the CFP and YFP protein with a short linker, which is used as a positive control for the FRET analysis. CFP/YFP indicates the co-expression of individual CFP and YFP protein, which is used as a negative control for the FRET analysis. Acceptor photobleaching FRET analysis was conducted on the Leica SP8 confocal system. Briefly, cells expressing various CFP and YFP fusions were used for photobleaching using 514 nm laser in full power intensity. CFP donor fluorescence was imaged before and after bleaching a region of interest of YFP to less than 10% of its initial intensity. FRET efficiency was calculated as Ef =100 × (IPost − Ipre)/Ipost, where Ipre and Ipost stand for the CFP intensities before and after acceptor bleaching, respectively.

## Particle bombardment of *Arabidopsis* seedlings

Five-day-old *Arabidopsis* seedlings were carefully transferred to a new MS plate containing 10 μM Dexamethasone in ordered arrangement for 24 h. Gold particles coated with plasmid DNA were prepared and bombarded three times at three different positions into the plants using PSD-1000/He particle delivery system (Gas pressure: 1100 psi; Vacuum: 28 mm Hg; Gap distance between the rupture disk and macrocarrier: 1 cm; Gold particle flight distance between macrocarrier and plant samples: 9 cm) as described previously[79,80]. Plants were kept in dark in the plant growth chamber for 12–24 h before confocal observation.

## High pressure freezing and transmission electron microscopy

5 DAG *Arabidopsis* seedling root tips were cut and high-pressure freezing/frozen by the high-pressure freezer (EM PACT2, Leica, Germany) prior to subsequent freeze substitution in acetone containing 0.4% uranyl acetate at −85 °C in an AFS freeze-substitution unit (Leica, Wetzlar, Germany). After gradient infiltration with increasing concentration of HM20, samples were embedded and ultraviolet polymerized for ultra-thin sectioning[48,78]. TEM analysis was performed by observing the ultra-thin section on the grids under the 80 kV Hitachi H-7650 transmission electron microscope (Hitachi High-Technologies Corporation, Japan) with a charge-coupled device (CCD) camera.

## 3D electron tomography

The electron tomography observation procedures have been described previously[81]. In general, 300-nm-thick sections were observed using a 200 kV Tecnai F20 electron microscope (FEI Company). For each grid, a tilt image stack from at least +51° to −51° with 1.5° increments was collected, while the second axis of tilt image stack was collected by rotating the grid by 90°. Dual-axis tomograms were calculated from pairs of image stacks with the etomo program of the IMOD software package. The 15 nm gold particles at the surface of the sections were used as fiducial markers to align individual images in the tilt series. For model generation, the contours were drawn manually and meshed with the 3dmod program in the IMOD software package.

## Proteomics and mass-spectrometry

Proteins were extracted from 5 DAG seedlings in IP buffer (50 mM Tris-HCl, pH 7.4, 150 mM NaCl, 1 mM MgCl₂, 20% glycerol, 0.2% NP-0.4, 1× protease inhibitor cocktail, Roche). The collected supernatant was then co-incubated with GFP-Trap magnetic beads (Chromotek) at 4 °C as instructed by the manufacture handbook. The beads were washed

five times with the same ice-cold IP buffer and eluted by boiling in reducing SDS sample buffer before SDS-PAGE gel electrophoresis. The gel was then stained with Imperial™ Protein Stain after SDS–PAGE separation, and proteins were cut out for in-gel trypsin digestion as previously described[82]. In general, the cut gel was dehydrated and rehydrated using acetonitrile and water repeatedly for three times before reducing the disulfide-bond using dithiothreitol (DTT) and iodizing the gel using iodoacetamide (IAA). The gel was further dehydrated using acetonitrile for three times before trypsin digestion at 30 °C overnight. The digested peptide mixture was fractionated by nano-liquid chromatography-MS (nano-LCMS) using an UltiMate 3000 RSLCnano system (Thermo Fisher Scientific). MS data were then acquired with an Orbitrap Fusion Lumus mass spectrometer (Thermo Fisher Scientific). For the phosphorylation sites identification, raw MS/MS data were converted to Mascot generic format (mgf) and were used to search database using an in-house Mascot search engine (Matrix Science) with phosphorylation. For the interactors identification, raw MS/MS data was analyzed with MaxQuant version 2.0.3.1 as described in previous study[55]. The reference sequences of *Arabidopsis thaliana* proteome (Araport11_pep_20220103.fasta) were from TAIR database (arabidopsis.org), which contain 48231 entries. The false discovery rate (FDR) threshold of peptide-to-spectrum match (PSM) and protein were set as 0.05. "Trypsin digestion" and "Orbitrap" were chosen in the parameters setting while other searching parameters were kept as default. The "iBAQ" value was set for label free quantification. Data matrix of protein expression levels in all samples were merged by in-house Perl scripts. The expression value was transformed through $\log_2(iBAQ+1)$. Differential expression analyses were performed by using *limma* package. We adopted the default method in *Limma* which uses an Empirical Bayes estimate to "moderate" the standard deviation in the *t*-test denominator using the distribution of all the standard deviations (Moderated paired *t*-test). P-values were adjusted through Benjamini-Hochberg's method for multiple comparisons (https://bioconductor.org/packages/release/bioc/vignettes/limma/inst/doc/usersguide.pdf). The volcano plot was generated by using *ggplot2* package in R.

## Yeast two-hybrid

For Y2H analysis, the corresponding PCR products were cloned into the pGBKT7 or pGADT7 vectors (Clontech) (Supplementary Data 2). Plasmids of each pair were co-transformed into the yeast AH109 strain. Transformants were first screened on synthetic drop-out medium lacking Trp and Leu. Positive colonies were further selected on SD medium lacking His, Trp, Leu with 3AT, or His, Trp, Leu and Ade. The experiments were performed independently for at least twice with similar results obtained. Due to the high self-activation activity of full-length FREE1 protein on BD vector, the C-terminal part of FREE1(231–601) was used for the yeast two hybrid assay as previously described[63].

## Immunoblotting and co-immunoprecipitation

Seedlings were ground in ice-cold extraction buffer (50 mM Tris-HCl, pH 7.4, 150 mM NaCl, 20% glycerol, 1% Triton, 1× protease inhibitor cocktail, Roche) for protein extraction. Cell debris was removed by centrifugation at 700 g then 16,000 g at 4 °C. The collected supernatant was then denatured by boiling in reducing SDS sample buffer. Extracted proteins were then applied to gel electrophoresis on 12% SDS-polyacrylamide gel electrophoresis (SDS-PAGE) gels. Immunoblot analysis was performed as previously described[78]. For co-immunoprecipitation (co-IP) analysis, transfected protoplasts or transgenic plants were homogenized in IP buffer (50 mM Tris-HCl, pH 7.4, 150 mM NaCl, 1 mM MgCl₂, 20% glycerol, 0.2% NP-0.4, 1× protease inhibitor cocktail, Roche). Lysates were centrifuged at 16,000 g for 4 °C and were incubated with GFP-Trap magnetic beads (ChromoTek) for 2–4 h at 4 °C. After incubation, the beads were washed five times

with ice-cold washing buffer (50 mM Tris-HCl, pH 7.4, 150 mM NaCl, 1 mM MgCl₂, 20% glycerol and 0.2% NP-40) and eluted by boiling in reducing SDS sample buffer. Samples were then separated by SDS–PAGE and analysed by immunoblot. Primary antibodies used in this study were anti-FREE1 (homemade), anti-NBR1 (AS142805), anti-ATG8 (Agrisera, AS142769), anti-GFP (Chromotek, 029762), anti-RFP (Abcam, ab125244), anti-HA (Abcam, ab18181), anti-MYC (Santa Cruz, SC789), and anti-SnRK1α1 (Agrisera, AS10919).

### In vivo phosphorylation assay

In vivo phosphorylation assay was performed using protoplasts prepared from transgenic plants expressing KIN10-HA upon autophagy induction. GFP-FREE1 or GFP-FREE1$^{S530A}$ were transiently expressed in protoplasts derived from KIN10-HA transgenic plants upon nitrogen starvation for at least 18 h. GFP-FREE1 or GFP-FREE1$^{S530A}$ were further enriched by immunoprecipitation with anti-GFP for further detection by Phos-tag biotin.

### Semi-in vitro phosphorylation assay

Semi-in vitro phosphorylation assay was carried out essentially according to the methods described previously[63]. His-Sumo-FREE1, His-Sumo-FREE1$^{S530AS533A}$, His-Sumo-FREE1$^{S530A}$ fusion proteins were expressed and purified from *E. coil* (Rosseta) with Ni-NTA Resin. HA or GFP-tagged KIN10 proteins were obtained by immunoprecipitation from UBQ::KIN10-HA, UBQ::KIN10-GFP transgenic plants. GFP proteins were obtained by immunoprecipitation from UBQ::GFP transgenic plants. For in vitro kinase assays, immunoprecipitated KIN10-HA, KIN10-GFP, or GFP were incubated with purified His-Sumo-FREE1, His-Sumo-FREE1$^{S530A}$ or His-Sumo-FREE1$^{S530AS533A}$ in 30 μl kinase buffer (10 mM Tris/HCl, pH 7.8, 10 mM MgCl₂, 0.5 mM dithiothreitol, 2 mM MnCl₂ and 25 mM ATP) for 1 h at room temperature. Samples were separated by SDS-PAGE and transferred to PVDF membrane followed by probing with Phos-tag BTL-111. To observe the total protein amount of His-Sumo fusions and KIN10 proteins used for Phos-tag detection, same amount of protein from each sample was separated by SDS-PAGE followed by either coomassie brilliant blue staining or western blotting analysis with anti-GFP or anti-HA antibody.

### In vitro phosphorylation assay

All GST- or His- fusion protein constructs were transformed into *Escherichia coli* BL21 cells and recombinant proteins were induced by 0.5 mM isopropyl-β-D-thiogalactopyranoside (IPTG). After over-night incubation at 16 °C, cells were collected, lysed, sonicated and centrifuged, and recombination proteins (including KIN10, SnAK2, and FREE1) were purified according to the manufacturer's protocol (GE Healthcare Life Science and MERCK). In vitro kinase assays were performed as previously described[83]. Each reaction mixture contained 0.1 mL [γ-$^{32}$P] ATP (1 μCi) in kinase buffer (20 mM Tris-HCl, pH 8.0, 5 mM MgCl₂, 10 mM [$^{31}$P] ATP, and 1 mM DTT). The kinase assays were performed at 30 °C for 30 min, and then were incubated with 6X SDS loading buffer at 100 °C for 10 min. The samples were separated by 10% SDS-PAGE gels and stained with Coomassie brilliant blue R250 (CBB), and then the gels were exposed to phosphor screens. The autoradiograph (Autorad) signal was detected by a Typhoon 9410 phosphor imager.

### Structure-illumination microscopy

Transfected *Arabidopsis* protoplasts samples were imaged on a Nikon NSIM system consisting of a Ti-E stand with Perfect Focus, CFI SR HP Apochromat TIRF 100XC Oil lens, Nikon LU5 laser bed (488 and 561 nm laser lines) and Andor DU-897X-5254 EMCCD camera. For each focal plane, 15 images (5 phases, 3 angles) were captured with the NIS Elements v4.1. SIM images were reconstructed using slice reconstruction in NIS elements. Post-imaging analysis was performed using NIS Elements v4.1.

### Molecular evolutionary genetics analysis

The evolutionary history was inferred using the Neighbor-Joining method. The optimal tree with the sum of branch length=4.32642842 is shown. The tree is drawn to scale, with branch lengths in the same units as those of the evolutionary distances used to infer the phylogenetic tree. The evolutionary distances were computed using the *p*-distance method and are in the units of the number of amino acid differences per site. The analysis involved 80 amino acid sequences. All positions with less than 95% site coverage were eliminated. That is, fewer than 5% alignment gaps, missing data, and ambiguous bases were allowed at any position. There were a total of 114 positions in the final dataset. Evolutionary analyses were conducted in MEGA7[84].

### Protease protection assay

Protease protection assay was performed as recently described (Supplementary Fig. 5a) with modifications[54]. Roots from seven-day-old *Arabidopsis* seedlings treated with or without DEX/BTH/ConcA were resected and grinded in GTEN-based buffer (GTEN (10% glycerol, 30 mM Tris (pH 7.5), 150 mM NaCl, 1 mM EDTA (pH 8)), 0.4 M sorbitol, 5 mM MgCl₂, 1 mM Dithiothreitol (DTT), protease inhibitor cocktail (SIGMA)) in a 3:1 v/w ratio. Lysates were then subjected to low-speed (10 min, 700 x *g*), high-speed (15 min, 12,000 x *g*), and ultra-speed (60 min, 100,000 x *g*) centrifugation to enrich the microsomal fraction containing autophagosomal membranes. The enriched autophagosomal membranes were proceeded for proteinase K digestions (30 ng/μl, 30 min), followed by protein extraction and subsequent western blot detection by NBR1 and Atg8 antibodies.

### Quantification and statistical analysis

Confocal quantification: For the autophagosome number quantification in *Arabidopsis* root cells, the ATG8e-positive puncta in the mid-optical section of each root cell were counted, at least 20 individual root cells were used for quantification analysis. For the unclosed autophagosome quantification in *Arabidopsis* root cells, the unclosed ATG8e-positive puncta in the mid-optical section of each root cell were counted, at least 20 individual root cells were used for quantification analysis. For the FRET assay using *Arabidopsis* PSBD protoplasts, over 50 puncta from at least 10 individual protoplasts were used for quantification analysis.

TEM quantification: for the quantification of unclosed autophagosome numbers in *Arabidopsis* root cells, the unclosed APs in the TEM thin section of each root cell were counted and over 20 individual root cells from at least 3 individual high-pressure freezing/frozen substituted *Arabidopsis* roots were used for quantification analysis.

All the data are presented as mean values±SD. Statistical analysis was performed by individual test as indicated using GraphPad Prism version 9.00 for mac (GraphPad Software Inc.). Differences in means were considered statistically significant at $^{*}P < 0.05$. Significance levels are: $^{*}P < 0.05$; $^{**}P < 0.01$; $^{***}P < 0.001$.

### Reporting summary

Further information on research design is available in the Nature Portfolio Reporting Summary linked to this article.

## Data availability

The authors declare that the data supporting the findings of this study are available in the paper and its supplementary information files. The raw data are available from the corresponding author upon reasonable request. The proteomic data generated in this study have been deposited in ProteomeXchange partner repository under accession code PXD040435. Source data are provided with this paper. The source data for Figs. 1–4, Supplementary Figs. 1, 2, 4–8, 11, 13–19, 21, 23–31 are provided as a Source Data file with this paper. Source data are provided with this paper.

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

## Acknowledgements

We thank Prof. Marisa S Otegui from University of Wisconsin-Madison for sharing the SNF7 antibody. This work was supported by grants from the National Natural Science Foundation of China (91854201), the Research Grants Council of Hong Kong (AoE/M-05/12, CUHK 14101219, C4033-19E, C4002-20W, C4002-21EF, C2009-19G, C4041-18E, C2003-22WF, R4005-18, and Senior Research Fellow Scheme SRFS2122-4S01), the Chinese University of Hong Kong (CUHK) Research Committee and CAS-Croucher Funding Scheme for Joint Laboratories to L.J.

## Author contributions

Y.Z. and L.J. conceived and designed research. Y.Z., B.L., S.H., H.L., W.C., Y.C., and G.L. performed experiments. Y.Z., W.C., Z.L., and J.G. generated the 3D-electron tomography models. B.L. and S.W.L. performed

proteomic and mass-spectrum analysis. H.L. and G.L. performed the in vitro and in vivo phosphorylation assay. B.L., S.H., Y.C., C.Y., J.Z., J.S., Y.G., C.G., and Y.D. generated resources. Y.Z., B.L., S.H., and L.F. analyzed data. Y.Z. wrote the paper. Y.Z., Y.D., and L.J. edited the paper. L.J. supervised the research.

## Competing interests

The authors declare no competing interests.
