## [Peer Review File · Nature Communications]

The Plant Unique ESCRT Component FREE1 Regulates Autophagosome ClosureReviewer #1 (Remarks to the Author):

In short, the work shows that FREE1 protein functions as a linkage between ATG conjugation system (specifically ATG8) and ECRTs to regulate the autophagosome closure under nutrient starvation. The authors also showed that FREE1 is activated by KIN10 mediated phosphorylation, evidencing that KIN10 not only activates autophagy by the direct phosphorylation of TOR (autophagy inhibitor) and ATG components, but also KIN10 is involved at later stages such as the sealing of autophagosomes.

Although there were isolated pieces of evidences pointing in this direction, this work has put together a great number of experiments to describe the involvement of FREE1 and its regulation in the process of autophagosome closure in plants. Therefore, I believe this work is original and substantially contributes to our current knowledge in this field.

The conclusions of this study are well supported by their data. For most claims, they use several independent approaches to support their statements (e.g. microscopy, immunoblots, etc.).

I could not pick up any substantial flaws along the manuscript. It is a well-thought study, well written, and it follows a nice logic.

The methodology is completely reasonable and well documented. In some cases, the identification and site mapping of protein phosphorylation by mass spectrometry would benefit the work, however this is not an impediment for publication. The work is already exhaustive, and the authors use mutagenesis of important residues to prove their relevance in activation/inactivation.

Minor comments:

To me there are some supplemental figures that need to be included in the manuscript. For instance, Supp. Fig 9c is way more relevant than Fig. 2d. Fig 2d is an isolated observation of an unsealed autophagosome in atg5 mutant, whereas Supp. Fig 9c is the outcome of analysing multiple microscopies evidencing that the number of unsealed autophagosomes are greater in atg5 mutant than in WT. I would at least include Fig 9c within Figure 2, and if I had to choose one I would select the bar plot instead of the microscopy.

In line 73, please provide references for the sentence.

In line 86, introduce ILV which is used later without introduction.

In Supp. Fig 15, the immunoblot for the co-IP samples seems to be at a lower position than it should. For instance, GFP looks lower than the 25 kDa mark.

The legends indicating the mutant lines used in Figure 5 b are very hard to read, and probably impossible in a printed version. I recommend to use a larger font size and/or write them on top of the figure b, rather than within the microscopy.

Reviewer #2 (Remarks to the Author):

The manuscript of Zeng et al provides interesting insights into the function of ESCRT machinery for autophagosome closure in Arabidopsis thaliana. It shows that FREE1, a plant-unique component of the ESCRT complex, is essential for the recruitment of ESCRT machinery on the autophagosome via interacting with some specific isoforms of ATG8. They also found that the phosphorylation of FREE1 by SnRK1 is proposed as another regulatory mechanism of FREE1 recruitment and autophagosome closure. The manuscript provides valuable insights into the final process for autophagosome

formation and the function of ESCRT machinery for it. I think the data are convincing, and that the manuscript is well written and illustrated. However, there are some major issues and ambiguities that need to be clarified.

Major comments

Line 184-244, 382:

The authors clearly showed that FREE1 interacts with some specific isoforms of ATG8 via a classical AIM motif of FREE1, and it is essential for the recruitment of FREE1 on the autophagosome. The authors also demonstrated that SNF7 and VPS4, components of ESCRTIII are localized on the autophagosome and essential for autophagosome closure. However, it is not unclear whether the interaction between FREE1 and ATG8 is required for the recruitment of other ESCRT machinery. In order to link these two important findings by the authors, I recommend observing the localization of the ESCRTIII component in the FREE1W438AI441A-expressing line.

Line 203-208, 217-221:

The authors found that FREE1 interacts with the ATG conjugation system. In order to understand the roles of the interaction, they analyzed autophagosome morphology and autophagy flux in the atg5-1 mutant. As the authors described, autophagosome formation is severely inhibited in the atg5-1 mutant. As far as I know, this is mainly due to the defects of ATG8 conjugation. Since the effects of ATG5 are likely to be diverse, it is difficult to conclude only from phenotypic observations of atg5 mutants that the relationship between FREE1 and the ATG conjugation machinery is important for autophagosome closure. Especially, the defect of autophagic flow in the atg5 mutant (line 217-221) is a very common phenotype of autophagy-deficient mutant, and it is impossible to conclude that autophagosome closure is the cause of it.

Line 212, 213, Fig. 3b:

The authors argue that the interaction between FREE1 and ATG8 is an evolutionarily conserved mechanism because FREE1 interacts with some specific ATG8 isoforms with weak phylogenetic relationships. However, it is not clear why FREE1 can only interact with some specific ATG8 isoforms. It is also unclear whether similar interaction specificity exists in plants other than Arabidopsis. Given these results, it is difficult to argue for evolutionary conservatism.

Line 306: It is difficult to understand why the authors experimented with snrk2.2/2.3 mutant, as I am not familiar with this field. I suppose the authors hypothesized that in the snrk2.2/2.3 mutant, the activity of SNRK1 will be enhanced. The authors should be clearly written the purposes and expectations.

Fig. 3e:

This is one of the key figures indicating the interaction between ATG8s and FREE1, but the band of 3Myc-FREE1 of Co-IP is unclear. Considering the beautiful image of a similar experiment shown in Supplementary Figure 11b, I recommend replacing a clearer one.

Minor comments

Line 197: ", HOPs, ..."

Abbreviations should be defined in the text or legends at their first occurrence. "HOPS" (not "HOPs") should be used as the abbreviation for "homotypic fusion and vacuole protein sorting".

Line 226: "Recent study has shown that ATG8 may interact with FREE1 via typical LDS, " LDS should be defined in the text. It should be clearly stated that this statement is based on the screening results performed in the cited paper (Marshall et al., 2019). The current text is confusing because of the unclear basis for the hypothesis and its relationship to the authors' results.

Line 273-325:

The authors have shown from multiple experiments that FREE1 is phosphorylated by

KIN10. However, the explanation is redundant and difficult to understand due to the many similar experiments. There are some parts where similar method descriptions are repeated, so why not rephrase or reduce those?

Line 470: "For DEX, BTH, or Concanamycin A (ConcA) treatments in CLSM observation, protein extraction analysis, and high-pressure freezing, 10 μ M DEX, 100 μ M BTH, and 1 μ M ConcA were applied to seedlings for indicated time points before analysis, respectively."

The method section describes the treatment times as above, but they are not properly described in the Figure legend.

Line 614: It should be described how to calculate FRET efficiency from the original fluorescent intensity. If you use any specific tool for it, it should be described here, too.

Fig. 1e, Supplementary Fig. 6: The meaning of the color difference in the 3D model should be described in the legend.

Fig. 2d (right panel): It is unclear for the open region. Is this the best plane to describe it?

Fig. 3a, 4c, Supplementary Fig. 8, 10, 11:

Explanations of SD-2 and SD-4 can be found only in the method section. It should be described at least in each Figure legend. I recommend using "-Trp -Leu" as an alternative to SD-2, and "-His -Ade -Trp -Leu" as an alternative to SD-4.

Fig. 4: This is very busy. Some of the images (Fig. 4j, k) are difficult to understand. It should be rearranged.

Fig. 6: This is also very busy. The diagram inside the circle is summarized their findings, but it is too small. Plant images at both sides can be smaller.

Supplementary Fig. 8a: This diagram is the truncated FREE1 for Y2H analysis. The full-length diagram should be shown like Supplementary Fig. 11a, and then the region used for Y2H should be illustrated.

Supplementary Fig. 18: In order to understand the positioning of the AIM motif, phosphorylation sites and mutation site of free1-ct, it should be described the diagram of FREE1 including all the above information.

Figure arrangements of Supplementary Figs:

There are 24 Supplementary Figures. Some of the Figures can be combined together (ex. Supplementary Figs. 14 and 15, and Supplementary Figs. 20-23).

Supplementary Movie 1 and 2: Both movies are not clear due to the low resolution to distinguish closed- or unclosed-form of autophagosomes. I recommend combining a higher magnified movie of one of the autophagosomes inserted at the corner of the movie. The time and scale bar should be indicated in the movies.

Reviewer #3 (Remarks to the Author):

In this manuscript, Zheng et al report the mechanism behind the regulation of autophagosome closure in plants. While the role of ESCRTs in autophagosome closure has been demonstrated in both yeast and human cells, how the fission machinery is regulated during the process remains unclear. The authors' group has previously identified the plant-specific ESCRT component FREE1 and demonstrated that the component plays dual roles in vacuolar protein transport and autophagic degradation. Here, they have performed ultrastructural analysis and found that the majority of the

autophagic structures accumulated in free1 mutants are unsealed. Mechanistically, they report that, upon the induction of autophagy, FREE1 is phosphorylated by SnRK1 to promote its recruitment to the autophagosomes for closure. The study is comprehensive and the findings may be potentially important for advancing our understanding of autophagosome closure. However, the methods designed to assess closure defect are not robust and occasionally inadequate (i.e. usage of ATG8 to monitor the closure in ATG5 mutants), and contain some inconsistencies between their own data. Moreover, there are several areas that require additional data to support their conclusions.

Major:

1. The 'unclosed' structures that the authors have claimed are found to be increased (about two-fold) by the V-ATPase inhibitor ConcA (Fig 1a,b, Supple Fig 13b,c) despite the impairment of ATG8-PE turnover/autophagic flux in FREE1 knockdown mutants (Suppl Fig 3) and VPS4DN-expressing seedlings (Suppl Fig 13d,e). As the similar 'unclosed' ATG8-positive structures can be detected in the absence of the conjugation system (although the number appears to be decreased, Supple Fig 10b), at least a portion of the fluorescent structures claimed as 'unclosed APs' could be membrane-less ATG8-positive inclusions rather than phagophores/autophagosomes. Immuno-EM and/or CLEM should be performed to determine the identity of the ATG8-positive structures.
2. Suppl Fig 5 show that nearly completed yet unclosed autophagosomes, but not 'cup-shaped' early phagophores and closed autophagosomes, are accumulated in free1 mutants. While these observations support the role of FREE1 in autophagosome closure, the data are in part contradictory to the images shown in Fig 1c in which all the 'unclosed AP' structures appear to be rather 'cup-shaped'. Are the structures shown in Fig 1c positive for ATG8? As Arabidopsis ATG1a and ATG13a are known to be degraded via autophagy (PMID: 21984698), the authors may determine if the unclosed structures accumulated in free1 mutants are also positive for these proteins.
3. More quantitative and reliable methods (i.e. protease protection assay, HaloTag-ATG8 assay if feasible) should be employed to demonstrate that FREE1 depletion indeed results in the accumulation of unsealed autophagosomes. As FREE1 appears to be well-localized with CFP-ATG5/12a/16 (note: the constructs should be validated; see comment 4), the authors may also examine if FREE1 knockdown accumulates these phagophore markers.
4. Are the ectopically expressed ATG5 and ATG12a constructs used in Fig 2b,c functional? Judging from the YFP blot in Fig 2c, all the ectopically expressed YFP-tagged ATG5 proteins are failed to be conjugated with endogenous ATG12. The co-IP experiments in Fig 2c may be conducted using anti-Myc to pulldown ectopically-expressed (or endogenous FREE1 using anti-FREE1) and determine if the ATG5 complex (ATG12-5a conjugate and ATG16) can be co-precipitated.
5. The data in Fig 3d, Suppl Fig 12b and Suppl 13a do not convincingly support the edge/rim localization of ESCRTs. Immuno-EM should be performed to demonstrate the claim. Alternatively, the authors may simply tone down the statement.
6. The effects of the ATG8 interaction-defective FREE1 mutations (W438/I441A and S530A) on ESCRT-III recruitment and autophagic flux (ATG8-PE and NBR1 turnover as well as YFP-ATG8e cleavage) should be determined. Do the mutations affect only autophagy but not vacuolar protein transport? Is the method used in Suppl 24 specific for monitoring autophagy?
7. Is ESCRT-I required for FREE1-mediated autophagosome closure?
8. It is unclear how important the FREE1 interaction with the ATG5 complex is for FREE1-dependent autophagosome closure. Supple Fig 11d,e and Fig 4m show that the ATG8 interaction-defective FREE1 mutants cannot localize on ATG8-labelled structures, some of which could be ATG5-positive unsealed autophagosomes. Do the FREE1 mutations affect its interaction with the ATG5 complex as well? If not, the authors may use ATG8 lipidation-defective ATG3 mutants to determine if FREE1 can translocate to the ATG5 complex-positive membranes independent of ATG8s.

Minor

9. Are the FREE1 constructs used in Suppl Fig 11 d,e resistant to FREE1 RNAi?

10. The dominant negative ESCRT mutants used in the study should be explained in the text.

11. Supple Fig 12a requires data quantification to state that BTH treatment significantly increased colocalization of ATG8e with SNF7.1. In addition, the experiment should be performed in the absence of ConCA.

12. The localization pattern of SNF7.1-GFP in Suppl Fig 12 looks to be abnormal. If the construct used in the experiment dominant negative?

Reviewer #4 (Remarks to the Author):

Zeng et al. describe in their paper the involvement of the plant unique ESCRT component FREE1 in autophagosome closure. The authors applied high-resolution microscopy, 3D-electron tomography, proteomic, cellular and biochemical approaches to elucidate the function of Arabidopsis FREE1. Specifically, during nutrient-deficient condition, SnRK1 is phosphorylating FREE1 that is subsequently recruited by the ATG conjugation system finally mediating, together with the downstream ESCRT machinery, the autophagosome closure. To my knowledge, these findings are completely new and highly relevant in the plant field as well as in the ESCRT and autophagy field.

The quality of writing is very good, the manuscript is clearly written, and the reader can easily follow the story. The authors applied highly efficient molecular and biochemical approaches; the microscopic analyses are high-end and the pictures of high quality. All methods are accurately planned and include all needed negative controls and amounts of samples and replicates for reliable statistical analyses. Taken together, this manuscript was a pleasure to read, the applied methods and number of controls impressive. Subsequently, the results and conclusions are accurate.

Thus, I highly recommend to publish this manuscript since I have only minor revisions: line 79: I think, this should be“on triggering the autophagosome initiation through phosphorylation (instead of phosphorylates) and activation (instead of activates) of....”

Line 109: please also indicate the reference Winter and Hauser (<https://doi.org/10.1016/j.tplants.2006.01.008>) since this paper point out the composition of the plant ESCRT machinery in 2006.

Line 606: please indicate the reference for MEGA7.

Concerning the methods:

.) can you please indicate in the Y2H assay which concentration of 3AT was used? Only Figure 4 indicates 8 mM 3AT

.) line 237: particle bombardment was mentioned but is not described in the methods part

Please check the consistency of VPS/Vps, Co-IP/co-IP/co-immunoprecipitation, Snf7/SNF7

REVIEWER COMMENTS

Reviewer #1 (Remarks to the Author):

In short, the work shows that FREE1 protein functions as a linkage between ATG conjugation system (specifically ATG8) and ESCRTs to regulate the autophagosome closure under nutrient starvation. The authors also showed that FREE1 is activated by KIN10 mediated phosphorylation, evidencing that KIN10 not only activates autophagy by the direct phosphorylation of TOR (autophagy inhibitor) and ATG components, but also KIN10 is involved at later stages such as the sealing of autophagosomes.

Although there were isolated pieces of evidences pointing in this direction, this work has put together a great number of experiments to describe the involvement of FREE1 and its regulation in the process of autophagosome closure in plants. Therefore, I believe this work is original and substantially contributes to our current knowledge in this field.

The conclusions of this study are well supported by their data. For most claims, they use several independent approaches to support their statements (e.g. microscopy, immunoblots, etc.).

I could not pick up any substantial flaws along the manuscript. It is a well-thought study, well written, and it follows a nice logic.

The methodology is completely reasonable and well documented. In some cases, the identification and site mapping of protein phosphorylation by mass spectrometry would benefit the work, however this is not an impediment for publication. The work is already exhaustive, and the authors use mutagenesis of important residues to prove their relevance in activation/inactivation.

Response: Thanks for your time and efforts in reviewing our manuscript.

Minor comments:

To me there are some supplemental figures that need to be included in the manuscript. For instance, Supp. Fig 9c is way more relevant than Fig. 2d. Fig 2d is an isolated observation of an unsealed autophagosome in *atg5* mutant, whereas Supp. Fig 9c is the outcome of analysing multiple microscopies evidencing that the number of unsealed autophagosomes are greater in *atg5* mutant than in WT. I would at least include Fig 9c within Figure 2, and if I had to choose one I would select the bar plot instead of the microscopy.

Response: We have now included the **original Supplementary Figure 9** (including the bar plot shown in **original Supplementary Figure 9c**) to the **original Figure 2** as the **new Figure 2d-f**, and moved the **original Figure 2d** showing an isolated observation of an unsealed autophagosome in *atg5* mutant as the **new Supplementary Figure 12**.

In line 73, please provide references for the sentence.

Response: We have now provided the reference (Signorelli et al., 2019, *Trends in Plant Sciences*) for the sentence in line 73 in the original Ms. (Line 73, Page 3)

In line 86, introduce ILV which is used later without introduction.

Response: We have now introduced ILV as “intraluminal vesicle (ILV) that formed inside the MVB by invagination and budding of membrane into the lumen of the endosome” in the revised Ms. (Lines 87-88, Page 3)

In Supp. Fig 15, the immunoblot for the co-IP samples seems to be at a lower position than it should. For instance, GFP looks lower than the 25 kDa mark.

Response: We have now carefully checked the original blot for the **original Supplementary Figure 15** and re-marked the size of the bands according to the original blot in **new Supplementary Figure 21b**. We have also included the original blot in the Supplementary Information showing the uncropped blot.

The legends indicating the mutant lines used in Figure 5 b are very hard to read, and probably impossible in a printed version. I recommend to use a larger font size and/or write them on top of the figure b, rather than within the microscopy.

Response: We have now written the legends indicating the mutant lines on top of the figures in **new Figure 5b**.

Reviewer #2 (Remarks to the Author):

The manuscript of Zeng et al provides interesting insights into the function of ESCRT machinery for autophagosome closure in *Arabidopsis thaliana*. It shows that FREE1, a plant-unique component of the ESCRT complex, is essential for the recruitment of ESCRT machinery on the autophagosome via interacting with some specific isoforms of ATG8. They also found that the phosphorylation of FREE1 by SnRK1 is proposed as another regulatory mechanism of FREE1 recruitment and autophagosome closure. The manuscript provides valuable insights into the final process for autophagosome formation and the function of ESCRT machinery for it. I think the data are convincing, and that the manuscript is well written and illustrated. However, there are some major issues and ambiguities that need to be clarified.

Response: Thanks for your time and efforts in reviewing our manuscript. We have now revised the manuscript with new experiments according to your suggestions as shown below.

Major comments

Line 184-244, 382:

The authors clearly showed that FREE1 interacts with some specific isoforms of ATG8 via a classical AIM motif of FREE1, and it is essential for the recruitment of FREE1 on the autophagosome. The authors also demonstrated that SNF7 and VPS4, components of ESCRTIII are localized on the autophagosome and essential for autophagosome closure. However, it is not unclear whether the interaction between FREE1 and ATG8 is required for the recruitment of other ESCRT machinery. In order to link these two important findings by the authors, I recommend observing the localization of the ESCRTIII component in the FREE1^{W438AI441A}-expressing line.

Response: To investigate whether the interaction between FREE1 and ATG8 is required for the recruitment of other ESCRT machinery including ESCRTIII components:

- (1) We have now performed the transient expression assay using protoplasts isolated from *Arabidopsis* PSBD suspension culture cells for confocal analysis. Indeed, our results showed that the ESCRT-III component SNF7.1-GFP failed to be recruited onto the ATG8-positive structures labelled by mCherry-ATG8e in the protoplasts expressing the FREE1 RNAi (**new Supplementary Figure 19c**). Similarly, SNF7.1-GFP failed to be recruited onto the ATG8-positive structures in the protoplasts expressing FREE1 RNAi and the AIM motif mutant Myc-FREE1^{W438AI441A} (**new Supplementary Figure 19c**). Nonetheless, the ESCRT-III component SNF7.1-GFP could be sufficiently recruited onto the ATG8-positive structures in the protoplasts expressing FREE1 RNAi and wild-type Myc-FREE1 (**new Supplementary Figure 19c**). Quantification analysis further supported the confocal observations (**new Supplementary Figure 19d**). Therefore, our results demonstrated that the AIM motif-dependent interaction between FREE1 and ATG8 is required for the recruitment of the ESCRT-III components onto the autophagosome.
- (2) We have also performed the immunogold-TEM analysis using the antibodies against endogenous plant ESCRT-III component SNF7 (gifted from Prof. Marisa Otegui; Buono et al., 2017, *Journal of Cell Biology*) in Col-0 wild-type plants and *free1* mutants. Our results showed that SNF7 can label the autophagosomal structures in Col-0 wild-type plants, but not the accumulated unclosed autophagosomal structures in *free1* mutants (**new**

Supplementary Figure 19a). Quantification analysis of gold particles on the autophagosomal structures further supported the TEM observations (**new Supplementary Figure 19b**). Therefore, our TEM results further proved that FREE1 is important for the recruitment of other ESCRT components including ESCRT-III onto the autophagosomes.

We have now included these new data (**new Supplementary Figure 19**) with proper descriptions in the revised Ms (Lines 289-306, Page 9).

Line 203-208, 217-221:

The authors found that FREE1 interacts with the ATG conjugation system. In order to understand the roles of the interaction, they analyzed autophagosome morphology and autophagy flux in the *atg5-1* mutant. As the authors described, autophagosome formation is severely inhibited in the *atg5-1* mutant. As far as I know, this is mainly due to the defects of ATG8 conjugation. Since the effects of ATG5 are likely to be diverse, it is difficult to conclude only from phenotypic observations of *atg5* mutants that the relationship between FREE1 and the ATG conjugation machinery is important for autophagosome closure. Especially, the defect of autophagic flow in the *atg5* mutant (line 217-221) is a very common phenotype of autophagy-deficient mutant, and it is impossible to conclude that autophagosome closure is the cause of it.

Response: We fully agreed that the effects of ATG5 are diverse in autophagosome formation besides its *bona fide* role in autophagosome closure (Mizushima et al., 2001, *Journal of Cell Biology*; Kishi-Itakura et al., 2014, *Journal of Cell Science*), and the defect of autophagic flow in *atg5* mutant is a common phenotype of the autophagy-deficient mutants.

- (1) To explore whether the relationship between FREE1 and the ATG conjugation machinery is important for the autophagosome closure, we have now investigated the potential sites on FREE1 essential for its interaction with the ATG conjugation system by performing the co-immunoprecipitation analysis as suggested by **Reviewer 3**. Notably, our results showed that ATG5 could interact with FREE1 but not FREE1^{S530A} in planta, suggesting that the S530 on FREE1 may be important for the interaction between FREE1 and ATG5 (**new Supplementary Figure 28**). We have thus further performed the lipidation assay and autophagic flux analysis to investigate *bona fide* role of S530 on FREE1 in autophagosome closure. Our results showed that the lipidated ATG8 and autophagic-receptor NBR1 were substantially accumulated while YFP-ATG8 turnover was significantly impeded in *DEX::RNAi-FREE1* mutant protoplasts overexpressing FREE1^{S530A}, suggesting that the S530 on FREE1, which is essential for the interaction with ATG5, is important for the autophagosome closure biochemically (**new Supplementary Figure 30**). Together with our previous results showing the *bona fide* role of S530 on FREE1 in regulating autophagosome sealing using 2D-TEM and 3D-tomography analysis (**new Figure 5**), our current results indicated that the interaction between FREE1 and ATG5, which is depending on S530, is important for autophagosome closure.
- (2) We agreed that it could be insufficient to conclude that autophagosome closure depends on ATG5 only from the defect of autophagic flow in the *atg5* mutant since this is a very common phenotype of autophagy-deficient mutant. Recently, the autophagic-related markers including STX17 (Tsuboyama et al., 2016, *Science*), ULK1 (Kishi-Itakura et al., 2014, *Journal of Cell Science*), and WIPI-1 (Kishi-Itakura et al., 2014, *Journal of Cell Science*) instead of ATG8 have been used to analyze the autophagosomal structures in the ATG5 mutants. We have thus tried to perform the confocal analysis with image

deconvolution on the ATG18a-GFP (close homolog of mammalian WIPI-1) in WT and *atg5-1* mutants. Our results showed that the ATG18a-GFP positive structures accumulated in *atg5-1* mutants are largely unclosed comparing with WT (**new Supplementary Figure 13a**). Quantification analysis further supported our confocal observations (**new Supplementary Figure 13b**). Consistent with the results in mammal, our results suggested that the unclosed autophagosomes accumulated in ATG5 mutants in plants.

We have now included these new data (**new Figure 5, new Supplementary Figure 13, new Supplementary Figure 28, and new Supplementary Figure 30**) with proper descriptions in the revised Ms (Lines 222-226, Page 7, Line 418, Page 12; Lines 419-422, Page 13; Lines 436-453, Page 13).

Line 212, 213, Fig. 3b:

The authors argue that the interaction between FREE1 and ATG8 is an evolutionarily conserved mechanism because FREE1 interacts with some specific ATG8 isoforms with weak phylogenetic relationships. However, it is not clear why FREE1 can only interact with some specific ATG8 isoforms. It is also unclear whether similar interaction specificity exists in plants other than *Arabidopsis*. Given these results, it is difficult to argue for evolutionary conservatism.

Response: Indeed, the mechanism underlying FREE1 interaction with specific ATG8 isoforms remains unknown, while it is also unclear whether similar interaction specificity exists in plants other than *Arabidopsis*. We agreed that it is hard to argue that interaction between FREE1 and ATG8 is an evolutionarily conserved mechanism at current stage, while it would be an interesting direction for future study in other plant species including crops, which is an ongoing project in our laboratory. We have thus now toned down our claim as “It would be of great interest in future study to reveal the mechanism underlying the recognition of FREE1 by specific ATG8s in regulating autophagosome closure and to answer whether the mechanism is evolutionary conserved in other plants.”

We have now included the new discussion as shown above in the revised Ms (Lines 486-488, Page 14).

Line 306: It is difficult to understand why the authors experimented with *snrk2.2/2.3* mutant, as I am not familiar with this field. I suppose the authors hypothesized that in the *snrk2.2/2.3* mutant, the activity of SNRK1 will be enhanced. The authors should be clearly written the purposes and expectations.

Response: We apologized for the unclear description of using the *snrk2.2/2.3* mutants in the original Ms. To clearly write the purposes and expectations of using *snrk2.2/2.3* mutants in the semi-*in vitro* phosphorylation experiment, we have now included an explanation as “FREE1 has been shown to be phosphorylated by SnRK2.2/2.3, promoting its translocation to the nuclei under ABA conditions (Li et al., 2019, *Nature Plants*), whereas a recent study has also shown the sequestration of SnRK1 by SnRK2-containing complexes to inhibit SnRK1 signaling (Belda-Palazon et al., 2020, *Nature Plants*). Therefore, to provide the direct evidence of FREE1 phosphorylation by SnRK1 but not by SnRK2.2/2.3 in the semi-*in vivo* phosphorylation assay, we also used the *Arabidopsis* protoplasts isolated from the *snrk2.2/2.3* mutants for the assay.”

We have now included a new description as shown above in the revised Ms (Lines 359-364, Page 11).

Fig. 3e: This is one of the key figures indicating the interaction between ATG8s and FREE1, but the band of 3Myc-FREE1 of Co-IP is unclear. Considering the beautiful image of a similar experiment shown in Supplementary Figure 11b, I recommend replacing a clearer one.

Response: We have now re-performed the co-IP experiment analysing the interaction between ATG8s and FREE1. Consistent with the previous results, our results showed that FREE1 can interact with both ATG8a and ATG8i (**new Figure 3e**).

We have now replaced the original data with the better image in **new Figure 3e**.

Minor comments

Line 197: “, HOPs, ...”

Abbreviations should be defined in the text or legends at their first occurrence. “HOPS” (not “HOPs”) should be used as the abbreviation for “homotypic fusion and vacuole protein sorting”.

Response: We have now revised HOPs as HOPS and defined HOPS (homotypic fusion and vacuole protein sorting) in the revised Ms. (Lines 211-212, Page 7)

Line 226: “Recent study has shown that ATG8 may interact with FREE1 via typical LDS, “LDS should be defined in the text. It should be clearly stated that this statement is based on the screening results performed in the cited paper (Marshall et al., 2019). The current text is confusing because of the unclear basis for the hypothesis and its relationship to the authors' results.

Response: We have now: (1) defined the LDS (LIR docking site); and (2) revised the text as “Using Yeast-two hybrid screening, Marshall *et al* has shown that ATG8 may interact with FREE1 via typical LDS (LIR docking site), suggesting that FREE1 may contain the classical AIM motif.” to clearly stated that our statement is based on the screening results from Marshall *et al.*, 2019. (Lines 242-243, Page 7)

Line 273-325:

The authors have shown from multiple experiments that FREE1 is phosphorylated by KIN10. However, the explanation is redundant and difficult to understand due to the many similar experiments. There are some parts where similar method descriptions are repeated, so why not rephrase or reduce those?

Response: We have now reduced similar method descriptions such as “To provide the direct evidences of FREE1 phosphorylation by SnRK1 α 1, the semi-*in vivo* phosphorylation assays were performed using the KIN10 enriched from transgenic seedlings expressing KIN10-GFP in SnRK2s mutants *snrk2.2/2.3* background by immunoprecipitation with anti-GFP and used to phosphorylate FREE1, FREE1^{S530A} or FREE1^{S530AS533A} tagged with His-sumo, which was further detected by Phos-tag biotin.” in Lines 307-311, Page 9 in the original Ms.

Line 470: “For DEX, BTH, or Concanamycin A (ConcA) treatments in CLSM observation, protein extraction analysis, and high-pressure freezing, 10 μ M DEX, 100 μ M BTH, and 1 μ M ConcA were applied to seedlings for indicated time points before analysis, respectively.”

The method section describes the treatment times as above, but they are not properly described in the Figure legend.

Response: We have now properly described the treatment times in the Figure legend. (e.g. Main text: Line 1031, Page 31; Line 1040, Page 31; Line 1045, Page 31; Line 1053, Page 31; Line 1056, Page 31; Line 1062, Page 32; Lines 1103-1104, Page 33; Line 1130, Page 33; Line 1135, Page 34; Supplementary material: Line 9, Page 3; Line 11, Page 3; Line 4, Page 4; Line 4, Page 8; Line 5, Page 9; Line 3, Page 10; Line 3, Page 16; Line 5, Page 17; Line 4, Page 18; Line 4, Page 22; Line 9, Page 23; Line 16, Page 23; Line 19, Page 23; Line 4, Page 24; Line 9, Page 24; Lines 6-7, Page 25; Line 4, Page 26; Line 6, Page 28; Line 7, Page 29; Line 5, Page 30; Line 10, Page 32; Lines 8-9, Page 36; Line 1, Page 42; Line 9, Page 42; Line 19, Page 42)

Line 614: It should be described how to calculate FRET efficiency from the original fluorescent intensity. If you use any specific tool for it, it should be described here, too.

Response: We have now described how to calculate FRET efficiency as “Acceptor photobleaching FRET analysis was conducted on the Leica SP8 confocal system. Briefly, cells expressing various CFP and YFP fusions were used for photobleaching using 514 nm laser in full power intensity. CFP donor fluorescence was imaged before and after bleaching a region of interest (ROI) of YFP to less than 10% of its initial intensity. FRET efficiency was calculated as $E_f = 100 \times (I_{\text{Post}} - I_{\text{Pre}})/I_{\text{Post}}$, where I_{Pre} and I_{Post} stand for the CFP intensities before and after acceptor bleaching, respectively.” in methodology part in the revised Ms. (Lines 572-578, Page 17)

Fig. 1e, Supplementary Fig. 6: The meaning of the color difference in the 3D model should be described in the legend.

Response: We have now described the meaning of the color difference in the 3D model in the legends of **original Figure 1e (new Figure 1e)** and **original Supplementary Figure 6 (new Supplementary Figure 9)**. We have also described the meaning of the color in other figures containing 3D model (**new Figure 5b, new Supplementary Figure 12b, and new Supplementary Figure 17g**). (Lines 1047-1049, Page 31; Lines 1136-1138, Page 34; Lines 6-7, Page 12 in Supplementary Material; Line 6, Page 16 in Supplementary Material; Lines 22-23, Page 23 in Supplementary Material)

Fig. 2d (right panel): It is unclear for the open region. Is this the best plane to describe it?

Response: We have now used other plane in the 3D-tomography to clearly describe the open region pointed by the arrowhead in **original Figure 2d (new Supplementary Figure 12b)**.

Fig. 3a, 4c, Supplementary Fig. 8, 10, 11: Explanations of SD-2 and SD-4 can be found only in the method section. It should be described at least in each Figure legend. I recommend using “-Trp -Leu” as an alternative to SD-2, and “-His -Ade -Trp -Leu” as an alternative to SD-4.

Response: We have now revised “SD-2” and “SD-4” to “-Trp -Leu” and “-His -Ade -Trp -Leu” in the revised Figures. (**new Figure 3a, new Figure 4c, new Supplementary Fig 11b, 11d, 11e, new Supplementary Fig 14a, and new Supplementary Fig 16c**)

Fig. 4: This is very busy. Some of the images (Fig. 4j, k) are difficult to understand. It should be rearranged.

Response: We have now combined the **original Figure 4j, k** to the **original Supplementary Figure 19** as the **new Supplementary Figure 25**, since they are all showing the phenotypes of *free1* and *free1* phosphorylation site mutants upon carbon starvation. (**new Figure 4** and **new Supplementary Figure 25**)

Fig. 6: This is also very busy. The diagram inside the circle is summarized their findings, but it is too small. Plant images at both sides can be smaller.

Response: We have now enlarged the text size inside the circle of the model and reduced the size of plant images at both sides as suggested. (**new Figure 6**)

Supplementary Fig. 8a: This diagram is the truncated FREE1 for Y2H analysis. The full-length diagram should be shown like Supplementary Fig. 11a, and then the region used for Y2H should be illustrated.

Response: We have now included the diagram showing the full-length of FREE1 in the **new Supplementary Figure 11a** (**original Supplementary Figure 8a**) and illustrated the region used for Y2H. (**new Supplementary Figure 11a**)

Supplementary Fig. 18: In order to understand the positioning of the AIM motif, phosphorylation sites and mutation site of *free1-ct*, it should be described the diagram of FREE1 including all the above information.

Response: We have now included the diagram of FREE1 with the information of AIM motif, phosphorylation site and mutation site of *free1-ct* as suggested. (**new Supplementary Figure 24a**)

Figure arrangements of Supplementary Figs: There are 24 Supplementary Figures. Some of the Figures can be combined together (ex. Supplementary Figs. 14 and 15, and Supplementary Figs. 20-23).

Response: We have now combined the **original Supplementary Figure 14** and **15** as **new Supplementary Figure 21**, and the **original Supplementary Figure 20-22** as **new Supplementary Figure 26** as suggested. (**new Supplementary Figure 21** and **new Supplementary Figure 26**)

Supplementary Movie 1 and 2: Both movies are not clear due to the low resolution to distinguish closed- or unclosed-form of autophagosomes. I recommend combining a higher magnified movie of one of the autophagosomes inserted at the corner of the movie. The time and scale bar should be indicated in the movies.

Response: Done as suggested. (**new Supplementary Movie 1** and **new Supplementary Movie 2**)

Reviewer #3 (Remarks to the Author):

In this manuscript, Zheng et al report the mechanism behind the regulation of autophagosome closure in plants. While the role of ESCRTs in autophagosome closure has been demonstrated in both yeast and human cells, how the fission machinery is regulated during the process remains unclear. The authors' group has previously identified the plant-specific ESCRT component FREE1 and demonstrated that the component plays dual roles in vacuolar protein transport and autophagic degradation. Here, they have performed ultrastructural analysis and found that the majority of the autophagic structures accumulated in *free1* mutants are unsealed. Mechanistically, they report that, upon the induction of autophagy, FREE1 is phosphorylated by SnRK1 to promote its recruitment to the autophagosomes for closure. The study is comprehensive and the findings may be potentially important for advancing our understanding of autophagosome closure. However, the methods designed to assess closure defect are not robust and occasionally inadequate (i.e. usage of ATG8 to monitor the closure in ATG5 mutants), and contain some inconsistencies between their own data. Moreover, there are several areas that require additional data to support their conclusions.

Response: Thanks for your time and efforts in reviewing our manuscript. In the revised MS, we have tried our best to fully address your concerns by performing additional experiments, revising main text, and providing more discussions as shown below.

Major:

1. The 'unclosed' structures that the authors have claimed are found to be increased (about two-fold) by the V-ATPase inhibitor ConcA (Fig 1a,b, Supple Fig 13b,c) despite the impairment of ATG8-PE turnover/autophagic flux in FREE1 knockdown mutants (Suppl Fig 3) and VPS4DN-expressing seedlings (Suppl Fig 13d,e). As the similar 'unclosed' ATG8-positive structures can be detected in the absence of the conjugation system (although the number appears to be decreased, Supple Fig 10b), at least a portion of the fluorescent structures claimed as 'unclosed APs' could be membrane-less ATG8-positive inclusions rather than phagophores/autophagosomes. Immuno-EM and/or CLEM should be performed to determine the identity of the ATG8-positive structures.

Response:

- (1) To address the concerns on whether at least a portion of ATG8-positive structures could be the membrane-less ATG8-positive inclusions rather than phagophores/autophagosomes, we have now generated the transgenic plants expressing eYFP-ATG8 in *free1* mutants and performed the immunogold-TEM analysis using homemade (anti-rabbit) or commercial (anti-mouse) GFP antibodies. Our results showed that both GFP antibodies could substantially label the unclosed autophagosomal structures accumulated in cells under TEM, while the GFP antibodies labelling on other intracellular compartments or aggregates could be barely observed (**new Supplementary Figure 7a**). Gold particles quantification analysis further supported our observations (**new Supplementary Figure 7b**). Our results thus demonstrated that the ATG8-positive structures in *free1* mutants are largely unclosed autophagosomes in nature.
- (2) We have also tried to perform the immunogold-TEM analysis using the GFP antibodies in the transgenic plants expressing YFP-ATG8 in *atg5-1* mutants, but failed to detect the positive structures. Indeed, only few ATG8-positive structures could be observed as we stated in the original Ms. We fully agreed that the usage of ATG8 to monitor the

autophagosome closure in *atg5-1* mutants could be inadequate since the conjugation system mutants strongly affected the ATG8 lipidation. Recent studies have shown the accumulation of the unclosed autophagosomes in ATG5 mutants in mammal by TEM observation, demonstrating the *bona fide* role of the ATG conjugation system in autophagosome closure (Mizushima et al., 2001, *Journal of Cell Biology*; Kishi-Itakura et al., 2014, *Journal of Cell Science*). Consistently, we found that the unclosed membrane structures indeed accumulated substantially in *atg5-1* mutant plants under TEM as shown in the **original Supplementary Figure 9 (new Figure 2d, 2e, 2f)**. The autophagic-related markers including STX17 (Tsuboyama et al., 2016, *Science*), ULK1 (Kishi-Itakura et al., 2014, *Journal of Cell Science*), and WIPI-1 (Kishi-Itakura et al., 2014, *Journal of Cell Science*) instead of ATG8 have been used to analyze the autophagosomal structures in the ATG5 mutants. We have thus tried to perform the confocal analysis with image deconvolution on the ATG18a-GFP (close homolog of mammalian WIPI-1) in WT and *atg5-1* mutants. Our results showed that the ATG18a-GFP positive structures accumulated in *atg5-1* mutants are largely unclosed comparing with WT (**new Supplementary Figure 13a**). Quantification analysis further supported our confocal observations (**new Supplementary Figure 13b**). Consistent with the results in mammal, our results suggested that the unclosed autophagosomes accumulated in ATG5 mutants in plants.

We have thus removed the confocal data in the **original Supplementary Figure 10b** and included these new data (**new Supplementary Figure 7** and **new Supplementary Figure 13**), as well as new descriptions in the revised Ms (Lines 183-189, Page 6; Lines 222-226, Page 7).

2. Suppl Fig 5 show that nearly completed yet unclosed autophagosomes, but not ‘cup-shaped’ early phagophores and closed autophagosomes, are accumulated in *free1* mutants. While these observations support the role of FREE1 in autophagosome closure, the data are in part contradictory to the images shown in Fig 1c in which all the ‘unclosed AP’ structures appear to be rather ‘cup-shaped’. Are the structures shown in Fig 1c positive for ATG8? As *Arabidopsis* ATG1a and ATG13a are known to be degraded via autophagy (PMID: 21984698), the authors may determine if the unclosed structures accumulated in *free1* mutants are also positive for these proteins.

Response:

The discrepancy of TEM images shown in the **original Figure 1c** might be due to the thin section (80 nm per section) of the 2D-TEM which would not be sufficient for observing the intact autophagosomal structures. Therefore, some of the autophagosomal structures may seem to be “cup-shaped” in certain section in *free1* mutants. We have thus performed the serial-2D TEM as shown in **new Supplementary Figure 8 (original Supplementary Figure 5)** to differentiated the initiated phagophore and the unsealed autophagosome in *free1* mutants. To achieve this, we performed serial-sections and collected more than 10 serial thin sections (80 nm per section) for 2D-TEM analysis, which would be sufficient for observing the intact autophagosomal structures and for the quantification analysis. The obtained results showed that most of the autophagosomes accumulated in *free1* mutants were largely unsealed, although some of them seem to be initiated phagophores or closed autophagosomes at certain section (**new Supplementary Figure 8A and 8B**), which are in consistent with the 3D-TEM analysis. Quantification analysis further supported the observations (**new Supplementary Figure 8C and 8D**).

To address the concerns on whether the unclosed structures shown in the **original Figure 1c** are positive for ATG8, ATG1a, and ATG13a, we have now performed the immunogold-TEM analysis using the endogenous ATG8 (Product no AS14 2769), ATG1a (Product no AS19 4274), and ATG13a (Product no AS19 4279) antibodies purchased from Agrisera. Our results showed that the unclosed structures in *free1* mutants are indeed positive for the ATG8, ATG1a, and ATG13a (**new Supplementary Figure 6a**). Gold particles quantification analysis further supported our observations (**new Supplementary Figure 6b**). Our results thus demonstrated that the unclosed structures are autophagosomal structures in nature.

We have now included these new data (**new Supplementary Figure 6** and **new Supplementary Figure 8**) with proper descriptions in the revised Ms (Lines 182-183, Page 6).

3. More quantitative and reliable methods (i.e. protease protection assay, HaloTag-ATG8 assay if feasible) should be employed to demonstrate that FREE1 depletion indeed results in the accumulation of unsealed autophagosomes. As FREE1 appears to be well-colocalized with CFP-ATG5/12a/16 (note: the constructs should be validated; see comment 4), the authors may also examine if FREE1 knockdown accumulates these phagophore markers.

Response:

(1) To employ more quantitative and reliable methods to demonstrate that FREE1 depletion indeed results in the accumulation of unsealed autophagosomes, we have now performed the fractionation for autophagosomal membranes enrichment and protease protection assay according to the protocol recently developed in plants (**new Supplementary Figure 5a**) (Zhao et al., 2022, *Journal of Cell Biology*). Our results showed that the autophagic-receptor NBR1 and ATG8 substantially accumulated in the FREE1 depletion mutants upon DEX treatment (**new Supplementary Figure 5b**). Notably, the accumulated NBR1 and ATG8 were significantly degraded upon proteinase K digestion in the FREE1 depletion mutants (**new Supplementary Figure 5b**), suggesting that the autophagosomal structures are largely unclosed in FREE1 depletion mutants. Our results thus supported that the autophagosomes accumulated in FREE1 depletion mutants are unclosed biochemically.

We have now included these new data (**new Supplementary Figure 5**) with proper descriptions in the revised Ms (Lines 169-173, Page 5; Lines 174-176, Page 6).

(2) To examine if FREE1 knockdown accumulated the phagophore markers including ATG5/12a/16, we have purchased the antibodies against the endogenous ATG5 (AS15 3060) and ATG16 (AS19 4280) from Agrisera and used for the western blot analysis. Our results showed that ATG5 and ATG16 indeed accumulated in the FREE1 knockdown mutants comparing with that in the wild-type plants upon autophagy induction (**new Supplementary Figure 15a**). Quantification analysis further supported our observations (**new Supplementary Figure 15b**). Our results thus supported that the phagophore markers accumulated in FREE1 depletion mutants.

We have now included these new data (**new Supplementary Figure 15**) with proper descriptions in the revised Ms (Lines 237-238, Page 7).

4. Are the ectopically expressed ATG5 and ATG12a constructs used in Fig 2b,c functional? Judging from the YFP blot in Fig 2c, all the ectopically expressed YFP-tagged ATG5 proteins are failed to be conjugated with endogenous ATG12. The co-IP experiments in Fig 2c may be

conducted using anti-Myc to pulldown ectopically-expressed (or endogenous FREE1 using anti-FREE1) and determine if the ATG5 complex (ATG12-5a conjugate and ATG16) can be co-precipitated.

Response: Recent studies have shown the proper functionality of GFP-tagged ATG5 and ATG12 in plants (Le Bar et al., 2013, *Nature Communications*; Chung et al., 2010, *Plant Journal*). Nonetheless, to further consolidate the *bona fide* colocalization and interaction of FREE1 with the ATG conjugation system, we have now purchased the antibodies against the endogenous ATG5 (AS15 3060), ATG12b (AS19 4278), and ATG16 (AS19 4280) from Agrisera and used for both localization analysis and interaction assay. Our results showed that the endogenous ATG5, ATG12b, and ATG16 substantially colocalizes with FREE1 in *Arabidopsis* protoplasts (**new Figure 2b**), while the endogenous ATG5 complex (ATG12-5a conjugate and ATG16) can be significantly co-immunoprecipitated with FREE1 in plants (**new Figure 2c**). Our results thus demonstrated that FREE1 indeed colocalizes and interacts with ATG5 complex.

We have now replaced these new data (**new Figure 2b and 2c**) with proper descriptions in the revised Ms (Line 217, Page 7).

5. The data in Fig 3d, Suppl Fig 12b and Suppl 13a do not convincingly support the edge/rim localization of ESCRTs. Immuno-EM should be performed to demonstrate the claim. Alternatively, the authors may simply tone down the statement.

Response: As suggested, we have tried to perform the immunogold-TEM analysis to demonstrate the edge/rim localization of ESCRTs using the antibodies against endogenous plant ESCRT-III component SNF7 (gifted from Prof. Marisa Otegui; Buono et al., 2017, *Journal of Cell Biology*) in Col-0 wild-type plants. Our results showed that SNF7 can significantly label the autophagosomal structures in Col-0 wild-type plants (**new Supplementary Figure 19a**). Gold particles quantification analysis further confirmed the TEM observations (**new Supplementary Figure 19b**). Therefore, our results proved that ESCRT-III components localized on the autophagosomes. Although we could indeed observe the edge/rim localization of SNF7 in some autophagosomal structures (as shown in images below), we mainly observed the random distribution of SNF7 on the autophagosomal structures in the immunogold-TEM analysis (as shown in **new Supplementary Figure 19a**). We have thus toned down the statements in the revised Ms including (i) “Confocal and super-resolution

imaging showed that FREE1 colocalized with ATG8a and ATG8i at the phagophore labeled by ATG5”; (ii) “Further analysis using transient expression in *Arabidopsis* protoplasts demonstrated that SNF7.1 not only exhibited similar localization pattern as previously shown, but also localized at the phagophore”; (iii) “Consistently, another ESCRT-III component VPS20 also localized to the autophagosomes upon autophagic induction”; and (iv) “Consistent with this, our study confirmed that the plant ESCRT-III proteins SNF7.1 and VPS20 localize at the autophagosomal structures”.

We have now included these new data (**new Supplementary Figure 19a** and **19b**) with proper descriptions in the revised Ms (Lines 230-232, Page 7; Lines 272-275, Page 8; Lines 275-276, Page 8; Lines 471-473, Page 14; Lines 289-295, Page 9).

6. The effects of the ATG8 interaction-defective FREE1 mutations (W438/I441A and S530A) on ESCRT-III recruitment and autophagic flux (ATG8-PE and NBR1 turnover as well as YFP-ATG8e cleavage) should be determined. Do the mutations affect only autophagy but not vacuolar protein transport? Is the method used in Suppl 24 specific for monitoring autophagy?

Response:

(1) To investigate the effects of the ATG8 interaction-defective FREE1 mutations (W438/I441A and S530A) on ESCRT-III recruitment, we have now performed the transient expression assay using protoplasts isolated from *Arabidopsis* PSBD suspension culture cells. Our results showed that the ESCRTIII component SNF7.1-GFP failed to be recruited onto the ATG8-positive structures labelled by mCherry-ATG8e when co-expressing with FREE1 RNAi (**new Supplementary Figure 19c**). Similarly, SNF7.1-GFP failed to be recruited onto the ATG8-positive structures when co-expressing with FREE1 RNAi and the AIM motif mutant Myc-FREE1^{W438AI441A}, as well as co-expressing with FREE1 RNAi and Myc-FREE1^{S530A} (**new Supplementary Figure 19c**). Nonetheless, the ESCRT-III component SNF7.1-GFP could be recruited to the ATG8-positive structures when co-expressed with FREE1 RNAi and wild-type Myc-FREE1 (**new Supplementary Figure 19c**). Quantification analysis further supported the confocal observations (**new Supplementary Figure 19d**). Therefore, our results demonstrated that the interaction between FREE1 and ATG8 is required for the recruitment of the ESCRT-III components onto the autophagosome.

We have now included these new data (**new Supplementary Figure 19c** and **19d**) with proper descriptions in the revised Ms (Lines 295-306, Page 9).

(2) To investigate the effects of the ATG8 interaction-defective FREE1 mutations (W438/I441A and S530A) on autophagic flux (ATG8-PE and NBR1 turnover as well as YFP-ATG8e cleavage), we have now performed the lipidation assay, NBR1 turnover assay, and YFP-ATG8 vacuolar turnover assay using the protoplasts isolated from FREE1 RNAi mutants. Our results showed that (i) the lipidated ATG8 significantly accumulated in the *DEX::RNAi-FREE1* mutant protoplasts overexpressing Myc-FREE1^{S530A}, *DEX::RNAi-FREE1* mutant protoplasts overexpressing Myc-FREE1^{W438AI441A}, as well as *DEX::RNAi-FREE1* mutant protoplasts with or without BTH treatment, while the effects could be restored by overexpressing wild-type Myc-FREE1 in *DEX::RNAi-FREE1* mutant protoplasts (**new Supplementary Figure 30a**); (ii) the autophagy receptor NBR1 significantly accumulated in the *DEX::RNAi-FREE1* mutant protoplasts overexpressing Myc-FREE1^{S530A}, *DEX::RNAi-FREE1* mutant protoplasts overexpressing Myc-

FREE1^{W438AI441A}, as well as *DEX::RNAi-FREE1* mutant protoplasts with or without BTH treatment, while the defects could be recovered by overexpressing wild-type Myc-FREE1 in *DEX::RNAi-FREE1* mutant protoplasts (**new Supplementary Figure 30c**); and (iii) the YFP-ATG8e turnover was significantly impeded in the *DEX::RNAi-FREE1* mutant protoplasts overexpressing Myc-FREE1^{S530A}, *DEX::RNAi-FREE1* mutant protoplasts overexpressing Myc-FREE1^{W438AI441A}, as well as *DEX::RNAi-FREE1* mutant protoplasts with or without BTH treatment, while the defects could be complemented by overexpressing wild-type Myc-FREE1 in *DEX::RNAi-FREE1* mutant protoplasts (**new Supplementary Figure 30e**). Quantification analysis further supported our observations (**new Supplementary Figure 30b, 30d, and 30f**). Our results thus demonstrated that FREE1^{W438AI441A} and FREE1^{S530A} failed to complement the defective of FREE1 mutants on ATG8 lipidation, NBR1 turnover, and YFP-ATG8e turnover, suggesting that the ATG8 interaction-defective FREE1 mutations (W438/I441A and S530A) affected autophagosome closure biochemically.

We have now included these new data (**new Supplementary Figure 30**) with proper descriptions in the revised Ms (Lines 436-453, Page 13).

- (3) To investigate whether the FREE1 mutations (W438/I441A and S530A) affect only autophagy but not vacuolar protein transport, we have now performed the transient expression assay using the protoplasts isolated from *Arabidopsis* PSBD suspension culture cells. Confocal analysis results showed that the vacuolar transport of Aleurain-mRFP remains normal when co-expressing with FREE1 RNAi+Myc-FREE1, FREE1 RNAi+Myc-FREE1^{W438AI441A}, or FREE1 RNAi+Myc-FREE1^{S530A} (**new Supplementary Figure 31a**). Nonetheless, the vacuolar trafficking of Aleurain-mRFP was impeded when co-expressing with FREE1 RNAi (**new Supplementary Figure 31a**). Quantification analysis further confirmed the confocal observations (**new Supplementary Figure 31b**). To further consolidate our results, we have also performed the biochemical analysis and showed that the full length Aleurain-mRFP was accumulated in protoplasts when co-expressed with FREE1 RNAi (**new Supplementary Figure 31c**), while Aleurain-mRFP was mostly degraded to RFP core in vacuole when co-expressed with FREE1 RNAi+Myc-FREE1, FREE1 RNAi+Myc-FREE1^{W438AI441A}, or FREE1 RNAi+Myc-FREE1^{S530A} (**new Supplementary Figure 31c**). Therefore, our results demonstrated that the FREE1 mutations (W438/I441A and S530A) affect only autophagy but not vacuolar protein transport.

We have now included these new data (**new Supplementary Figure 31**) with proper descriptions in the revised Ms (Line 453, Page 13; Lines 454-456, Page 14).

- (4) The method used in **original Supplementary Figure 24** (**new Supplementary Figure 29**) is specific for monitoring autophagy and commonly used in plant autophagy research (Liu et al., 2012, *Plant Cell*). We have now included the proper citation in the revised Ms. (Line 427, Page 13).

7. Is ESCRT-I required for FREE1-mediated autophagosome closure?

Response: In *Arabidopsis*, all the ESCRT-I components VPS37, VPS23, and VPS28 contain two homologues encoding in the genome. Double mutant of the ESCRT-I components is inapplicable for cellular analysis due to the seedling lethality, while the single mutant does not exhibit obvious growth defective phenotypes under normal condition (Nagel et al., 2017,

PNAS). Recent study has also shown that neither the single mutant *vps23a* nor *vps23b* mutants showed autophagic flux defects (Zhao et al., 2022, *Journal of Cell Biology*). Nonetheless, to address the comment on whether ESCRT-I is required for FREE1-mediated autophagosome closure, we have tried to use the VPS23 single mutant for analysis. Our phenotypic analysis showed that neither the single mutant *vps23a* nor *vps23b* mutants exhibited comparable autophagy-related phenotypes as *atg5-1* and *atg7-2* upon carbon starvation (**new Supplementary Figure 20a**). TEM results showed that the autophagosomes closed normally in *vps23a*^{-/-}*vps23b*^{+/-} mutants, which is comparable with the wild-type plants (**new Supplementary Figure 20b**). Therefore, our current results suggested that VPS23A may either act redundantly with VPS23B in autophagosome closure or do not play role in autophagy. Due to the lethality of the double mutants of VPS23, future study could be carried out to generate micro-RNA mutants of both VPS23A and VPS23B driven by inducible promoter and used for exploring the possible functional roles of ESCRT-I in autophagosome closure.

We have now included these new data (**new Supplementary Figure 20**) with proper descriptions and a new discussion as shown above in the revised Ms (Lines 308-313, Page 9; Lines 314-321, Page 10).

8. It is unclear how important the FREE1 interaction with the ATG5 complex is for FREE1-dependent autophagosome closure. Supple Fig 11d,e and Fig 4m show that the ATG8 interaction-defective FREE1 mutants cannot localize on ATG8-labelled structures, some of which could be ATG5-positive unsealed autophagosomes. Do the FREE1 mutations affect its interaction with the ATG5 complex as well? If not, the authors may use ATG8 lipidation-defective ATG3 mutants to determine if FREE1 can translocate to the ATG5 complex-positive membranes independent of ATG8s.

Response: To address the concern whether FREE1 mutations affect its interaction with the ATG5 complex, we have now performed the transient expression and co-immunoprecipitation assay using protoplasts derived from *Arabidopsis* PSBD suspension culture cells. Our results showed that the FREE1 could interact with ATG5, while the mutations FREE1^{S530A} and FREE1^{W438AI441A} could not be co-immunoprecipitated with ATG5 (**new Supplementary Figure 28**). We thus demonstrated that W438I441 and S530 on FREE1 may be essential for forming a complex with ATG5 in plants.

We have now included these new data (**new Supplementary Figure 28**) with proper descriptions in the revised Ms (Line 418, Page 12; Lines 419-422, Page 13).

Minor

9. Are the FREE1 constructs used in Suppl Fig 11 d,e resistant to FREE1 RNAi?

Response: The FREE1 constructs used in **original Supplementary Figure 11** were not resistant to FREE1 RNAi. The FREE1 RNAi transgenic plants utilized the hairpin RNAi method to knock down the expression of FREE1 protein (Gao et al., 2014, *Current Biology*), which would be inapplicable to express the RNAi resistant constructs for the analysis. In mammal, studies have used the overexpression of the RNAi targeted genes to compensate the knock-down effects for protein function analysis in autophagy pathway (Li et al., 2021, *Cell Research*). We have thus performed the experiments in **new Supplementary Figure 16** (**original Supplementary Figure 11**) by inducing the knock-down system first, then overexpression of FREE1 wild-type and mutant variants to investigate whether the mutants fail to restore the autophagosomes closure using the particle bombardment. We have now further

performed western blot analysis and showed the proper overexpression of the FREE1 protein and its mutant variants in the FREE1 RNAi mutants (**new Supplementary Figure 19e** and **new Supplementary Figure 31c**).

We have now included these new data (**new Supplementary Figure 19e** and **new Supplementary Figure 31c**) with proper descriptions in the revised Ms (Lines 303-305, Page 9).

10. The dominant negative ESCRT mutants used in the study should be explained in the text.

Response: We have now explained the dominant negative ESCRT mutants used in this study as “we then generated dominant-negative mutant plants expressing SNF7.1(L22W) (SNF7.1DN), which has been shown to cause the enlargement of the PVC/MVB with a reduced number of intraluminal vesicles (ILVs) (Cai et al., 2014, *Plant Physiology*).” and “Consistently, dominant negative mutants of VPS4(E232Q) (VPS4DN), the ATPase-deficient form of VPS4 that causes the enlargement of the PVC/MVB with a reduced number of ILVs (Haas et al., 2007, *Plant Cell*)” in the revised Ms with proper citations (Cai et al., 2014, *Plant Physiology*; Haas et al., 2007, *Plant Cell*). (Line 278, Page 8; Line 279, Page 9; Lines 285-287, Page 9)

11. Supple Fig 12a requires data quantification to state that BTH treatment significantly increased colocalization of ATG8e with SNF7.1. In addition, the experiment should be performed in the absence of ConcA.

Response:

- (1) We have now included the data quantification for **original Supplementary Figure 12a** (**new Supplementary Figure 17a**) to state that BTH treatment significantly increased colocalization of ATG8e with SNF7.1. (**new Supplementary Figure 17b**)
- (2) We have now also performed the experiment for BTH treatment in the absence of ConcA. Our results showed the significantly increment of colocalization between ATG8e and SNF7.1 upon BTH treatment comparing with mock. (middle panel, **new Supplementary Figure 17a**)

12. The localization pattern of SNF7.1-GFP in Suppl Fig 12 looks to be abnormal. If the construct used in the experiment dominant negative?

Response: The construct used in the experiment is wild-type SNF7.1-GFP. Recent study has shown that SNF7.1-YFP exhibited puncta/aggregates/tonoplast pattern upon overexpression in *Arabidopsis* protoplasts (Cai et al., 2014, *Plant Physiology*), and our data are consistent with previous results showing the puncta as well as slightly tonoplast localization of SNF7.1 in *Arabidopsis* protoplasts system.

We have now included a brief description of such result with the reference (Cai et al., 2014, *Plant Physiology*) in the revised MS (Lines 272-275, Page 8; Lines 925-928, Page 27).

Reviewer #4 (Remarks to the Author):

Zeng et al. describe in their paper the involvement of the plant unique ESCRT component FREE1 in autophagosome closure. The authors applied high-resolution microscopy, 3D-electron tomography, proteomic, cellular and biochemical approaches to elucidate the function of *Arabidopsis* FREE1. Specifically, during nutrient-deficient condition, SnRK1 is phosphorylating FREE1 that is subsequently recruited by the ATG conjugation system finally mediating, together with the downstream ESCRT machinery, the autophagosome closure. To my knowledge, these findings are completely new and highly relevant in the plant field as well as in the ESCRT and autophagy field.

The quality of writing is very good, the manuscript is clearly written, and the reader can easily follow the story. The authors applied highly efficient molecular and biochemical approaches; the microscopic analyses are high-end and the pictures of high quality. All methods are accurately planned and include all needed negative controls and amounts of samples and replicates for reliable statistical analyses. Taken together, this manuscript was a pleasure to read, the applied methods and number of controls impressive. Subsequently, the results and conclusions are accurate.

Thus, I highly recommend to publish this manuscript since I have only minor revisions:

Response: Thanks for your time and efforts in reviewing our manuscript.

line 79: I think, this should be”on triggering the autophagosome initiation through phosphorylation (instead of phosphorylates) and activation (instead of activates) of....”

Response: Done. (Line 79, Page 3; Line 329, Page 10)

Line 109: please also indicate the reference Winter and Hauser (<https://doi.org/10.1016/j.tplants.2006.01.008>) since this paper point out the composition of the plant ESCRT machinery in 2006.

Response: We have now included the reference as suggested. (Lines 852-853, Page 25)

Line 606: please indicate the reference for MEGA7.

Response: We have now included the reference for MEGA7. (Line 707, Page 21; Lines 1007-1008, Page 30)

Concerning the methods:

.) can you please indicate in the Y2H assay which concentration of 3AT was used? Only Figure 4 indicates 8 mM 3AT

Response: Except using SD-3 with 3AT in the Y2H assay in Figure 4, Y2H assays in other Figures used SD-4 without 3AT. For better understanding, we have also revised “SD-2”, “SD-2” and “SD-4” to “-Trp -Leu”, “-His -Trp -Leu” and “-His -Ade -Trp -Leu” in the revised Figures as also suggested by Reviewer 2. (new Figure 3a, new Figure 4c, new Supplementary Fig 11b, 11d, 11e, new Supplementary Fig 14a, and new Supplementary Fig 16c)

.) line 237: particle bombardment was mentioned but is not described in the methods part
Please check the consistency of VPS/Vps, Co-IP/co-IP/co-immunoprecipitation, Snf7/SNF7

Response: We have now included the method of particle bombardment. (Lines 580-585, Page 17) We have also checked the consistency of VPS/Vps, Co-IP/co-IP/co-immunoprecipitation, Snf7/SNF7 in the revised Ms. (e.g. Line 95, Page 3; Line 103, Page 3; Line 104, Page 4; Line 265, Page 8; Line 232, Page 7; Line 644, Page 19)

Reviewer #1 (Remarks to the Author):

All my minor concerns were addressed and I have no further comments to add. The paper is of particular interest in our area, provides a clear advance in the understanding of autophagosome closure, and conclusions are supported by multiple methodological approaches.

S. Signorelli

Reviewer #2 (Remarks to the Author):

As with the original submission, I still feel that the study provides valuable insights into the detailed function of ESCRT machinery for autophagosome closure in *Arabidopsis thaliana*.

The authors have satisfactorily addressed all the comments I made. I have no further comments to make.

Reviewer #3 (Remarks to the Author):

The authors have addressed most of my concerns. However, I still have questions on the data in Fig 1a&b, which show a significant (about 2-fold) increase of BTH-induced YFP-ATG8e-positive "unclosed" autophagic structures by ConCA. If I understand correctly, the authors have interpreted this as an increase in autophagic flux (line 157-158). It should be stressed that ConCA has been widely used to not enhance but instead block autophagic flux; the ref 50 and 51 as well as the author's previous study (PMID: 25624505) also used the V-ATPase inhibitor to block autophagic/vacuole degradation. Does free1 loss only partially block autophagosome closure? Alternatively, or in addition, are the majority of unclosed autophagosomes in free1 mutants still subjected to vacuole fusion and degradation upon induced conditions? Adding DEX+BTH and DEX+BTH+ConCA treatments in Fig 5S might help to address these questions. I feel that these must be clarified in prior to publication in Nature Communications to avoid misleading.

REVIEWER COMMENTS

Reviewer #1 (Remarks to the Author):

All my minor concerns were addressed and I have no further comments to add.

The paper is of particular interest in our area, provides a clear advance in the understanding of autophagosome closure, and conclusions are supported by multiple methodological approaches.

S. Signorelli

Response: Thanks again for your time and efforts in reviewing our manuscript.

Reviewer #2 (Remarks to the Author):

As with the original submission, I still feel that the study provides valuable insights into the detailed function of ESCRT machinery for autophagosome closure in *Arabidopsis thaliana*.

The authors have satisfactorily addressed all the comments I made. I have no further comments to make.

Response: Thanks again for your time and efforts in reviewing our manuscript.

Reviewer #3 (Remarks to the Author):

The authors have addressed most of my concerns. However, I still have questions on the data in Fig 1a&b, which show a significant (about 2-fold) increase of BTH-induced YFP-ATG8e-positive “unclosed” autophagic structures by ConcA. If I understand correctly, the authors have interpreted this as an increase in autophagic flux (line 157-158). It should be stressed that ConcA has been widely used to not enhance but instead block autophagic flux; the ref 50 and 51 as well as the author’s previous study (PMID: 25624505) also used the V-ATPase inhibitor to block autophagic/vacuole degradation. Does *free1* loss only partially block autophagosome closure? Alternatively, or in addition, are the majority of unclosed autophagosomes in *free1* mutants still subjected to vacuole fusion and degradation upon induced conditions? Adding DEX+BTH and DEX+BTH+ConcA treatments in Fig 5S might help to address these questions.

I feel that these must be clarified in prior to publication in Nature Communications to avoid misleading.

Response: Thanks again for your time and efforts in reviewing our manuscript.

(1) In plants, recent study (Luo et al., 2017, *Frontier in Plant Science*) has shown that upon autophagy-induced condition such as salt stress (NaCl), additional ConcA treatment (+NaCl+ConcA) could not only induce the autophagic bodies accumulation in the central vacuole, but also further increase the number of autophagosomes in the cytoplasm by around two-folds comparing with salt treatment (+NaCl) only. Furthermore, Yano et al., (Yano et al., 2015, *Plant Signaling & Behavior*) has used the electron microscopy to reveal that not only the autophagic bodies were accumulated in the central vacuole of tobacco BY-2 cells, but also that autophagosome-like structures were more frequently observed in the cytoplasm in cells treated with ConcA comparing to the control, suggesting that the number of autophagosomes structures increases upon ConcA treatment in plants. These results suggested that, besides promoting the autophagic body accumulation in the central vacuole, ConcA treatment may also further induce the autophagosome formation in the cytoplasm upon autophagy-induced conditions, albeit the mechanism remains elusive. We have now included these descriptions with proper citations in the revised Ms. (Lines 161-164, Page 5)

(2) The concerns: “Does *free1* loss only partially block autophagosome closure? Alternatively, or in addition, are the majority of unclosed autophagosomes in *free1* mutants still subjected to vacuole fusion and degradation upon induced conditions?” Our previous study has shown that autophagic bodies could be barely observed in the vacuole of *free1* mutants comparing with the wild-type Col-0 upon BTH and ConcA treatments (Gao et al., 2015, *PNAS*). In the transgenic plants expressing EYFP-ATG8e (autophagy marker)/mCherry-VAMP711 (tonoplast marker) in *free1* T-DNA insertional or DEX-inducible mutant backgrounds, the EYFP-ATG8e puncta were largely absent from the vacuole, while the EYFP-ATG8e punctate structures were substantially accumulated in the vacuole of the wild-type Col-0 plants upon BTH and ConcA treatments (as shown below, left panel, extracted from **Figure 5** of Gao et al., 2015, *PNAS*). These results suggested that the loss of FREE1 in *free1* mutant strongly blocked the autophagosome closure and the unclosed autophagosomes in *free1* mutants could not be properly fused with the vacuole for

degradation upon induced conditions. We have now included these discussions with proper citations in the revised Ms. (Lines 495-500, Page 15)

- (3) Furthermore, we have now performed the suggested experiment by adding the DEX+BTH and DEX+BTH+ConcA treatments with new data shown in **Fig 5S**. Consistent with our confocal observations and the previously published data, western blot analysis showed more accumulation of both the autophagic-receptor NBR1 and ATG8 upon DEX+BTH+ConcA treatments comparing with the DEX+BTH treatments (**new Fig 5Sc**). We have now included these new data in **new Fig 5Sc** with proper description in the revised Ms. (Lines 181-183, Page 6)

Taken together, we agreed that the original interpretation of the BTH and ConcA treatments as an increase in autophagic flux may cause misleading, we have thus now rephrased our description as “Intriguingly, substantial amounts of unclosed autophagosomal structures were observed upon BTH-induced autophagy, while the effects were even more pronounced upon BTH and ConcA treatments (Fig. 1a, b). These results are consistent with the previous findings that the specific V-ATPase inhibitor ConcA which blocks autophagic/vacuole degradation may further induce the autophagosome formation in the cytoplasm upon autophagy-induced conditions, albeit the underlying mechanism remains elusive.” in the revised Ms. (Lines 161-164, Page 5)

(Extracted from **Figure 5** of Gao et al., 2015, *PNAS*)